



# Lava flow hazard modelling during the 2021 Fagradalsfjall eruption, Iceland: Applications of MrLavaLoba

Gro B. M. Pedersen[1], Melissa A. Pfeffer[2], Sara Barsotti[2], Simone Tarquini[3], Mattia de´ Michieli Vitturi[3,4], Bergrún Óladóttir[2], Ragnar Heiðar Þrasstarson[2]

[1]Nordic Volcanological Center, Institute of Earth Sciences, University of Iceland, Reykjavík, 102, Iceland
[2]Icelandic Meteorological Office, Reykjavík, 105, Iceland
[3]Istituto Nazionale di Geofisica e Vulcanologia, Pisa, 56127, Italy
[4]Department of Geology, University at Buffalo, Buffalo, New York 14260, USA

*Correspondence to*: Gro B. M. Pedersen (gro@hi.is)

## Abstract

On March 19, 2021, the first eruption in ca. 800 years took place in Fagradalsfjall on the Reykjanes Peninsula, in the backyard of the capital Reykjavík. This effusive eruption was the most visited eruption in Iceland to date and needed intense

lava flow hazard assessment and became a test case for hazard assessment for future eruptions on the Peninsula, which can issue lava into inhabited areas or inundate essential infrastructure.

In this study we documented how lava flow modelling strategies were implemented using the stochastic code MrLavaLoba, evaluating hazards during the 6-month long effusive event. Overall, the purposes were three-fold; (a) Pre-eruption simulation to investigate potential infrastructure at danger for lava flow inundation (b) Syn-eruptive simulations for short-term (two

weeks' time frame) lava flow hazard assessment and (c) Syn-eruptive simulations for long-term hazard assessments (months to years). Furthermore, strategies for lava barrier testing were developed and incorporation of near-real time syn-eruptive topographic models were implemented.

During the crisis the code was updated to increase functionalites such as considering multiple active vents as well as code

optimization that led to a substantial decrease in the computational time required for the simulations, speeding up the delivery of final products.

## 1 Introduction

On March 19, 2021, an eruption started at Mt. Fagradalsfjall on the Reykjanes Peninsula, Iceland, a mountainous area cut by nested enclosed valleys (Fig. 1). Being the first eruption on Reykjanes Peninsula in ca. 800 years and being located in the

backyard of the capital Reykjavík and the international airport, this eruption was the most visited eruption in Iceland to date.



It was easily accessible to the 2/3 of the Icelandic population as well as international tourists, and was visited by thousands of tourists per day, and hence the eruption needed intense monitoring and thorough hazard assessment (Barsotti et al., in review). Luckily, the 2021 Fagradalsfjall eruption did not impact any critical infrastructure. However, since several volcanic systems on the Reykjanes peninsula have the potential to issue lava into inhabited areas or inundate critical infrastructure,

the eruption became a test case for the monitoring and hazard assessment for future eruptions in the area that can be more destructive.

In this study we document how various lava flow modelling strategies using the stochastic code MrLavaLoba were implemented during the pre-eruptive unrest phase and during the eruption. The code proved to be a useful and a flexible tool

to evaluate pre-eruption as well as syn-eruptive short-term and long-term hazards during the 6-month long effusive event. Changes in approaches as well as new developments of the code were used to account for the changes in the eruptive behavior, and to resolve challenges provided by the complex topographic terrain, where infilling and overflowing of nested valleys created time-evolving hazards for visitors. Furthermore, strategies for lava barrier testing were developed and near-real time syn-eruptive topographic models were incorporated as the eruption progressed. In spite of recent technological

progresses, the so-called "deterministic" lava flow models tackling the physics of the lava emplacement provide only simplified solutions (e.g., the vertical structure of lava flows is typically not considered), at the cost of greater complexity and greater computational requirements. For this reason, we preferred to use the stochastic model MrLavaloba because it accounts for the lava flow volume and modify the topography during the simulated lava emplacement. In this work we also address caveats that should be considered when applying the code and make suggestions for future improvements to the

MrLavaLoba code.

### 1.1. Lava flow simulations

Lava flow modelling is a well-known tool to anticipate lava flow emplacement and lava flow models are commonly used for hazard and risk assessments before and during eruptions. Existing lava flow models are often divided into

deterministic codes and probabilistic (or stochastic) codes. The deterministic codes are intended to mimic the behavior of the natural systems by calculating physical processes based on a suite of physical properties (e.g., Dietterich et al., 2017, FLOWGO: Harris and Rowland, 2001, PyFLOWGO: Chevrel et al., 2018). Lava flows tend to follow the steepest path of decent downhill, but they do deviate from it in a probabilistic way, which is captured by stochastic codes (e.g., DOWNFLOW: Favalli et al., 2005, Tarquini and Favalli, 2013, Q-LAVHA: Mossoux et al., 2016). Recent developments of

probabilistic codes have included erupted volume as a parameter allowing the thickness of lava field to be estimated (Glaze and Baloga, 2013, MrLavaLoba: de' Michieli Vitturi and Tarquini 2018). The deterministic codes and some of the probabilistic ones attempt to replicate the patterns of channelized lava flows (e.g., Mossoux et al., 2016, Diettrich et al., 2017, Chevrel et al., 2018), while a few probabilistic codes additionally replicate tube-fed flows (Favalli et al., 2005, de' Michieli Vitturi and Tarquini, 2018). The probabilistic code "MrLavaLoba" can also account for the erupted volume and the



syn-eruptive modification of the topography (de' Michieli Vitturi and Tarquini, 2018).

Starting with the pioneering work at Mt. Etna during the 1991-93 eruption (Barberi and Villari, 1994), the numerical modeling of lava flows has been increasingly used to mitigate the destruction that can be caused by active lava flows. This approach has been refined over the years in both theoretical and practical respects (e.g., Wright et al., 2008, Cappello et al.,

2016, Harris et al., 2019).

Since 2007, the Hawaiian Volcano Observatory (HVO) has used the paths of steepest descent to assess likely lava flow routes during effusive crises (Kauahikaua, 2007). In a recent example, during the 2018 eruption in the Puna district (Neal et al., 2019), the HVO produced preliminary lava flow path forecasts using the DOWNFLOW code (Favalli et al. 2005). Later on, during the progression of the same effusive crisis, several lava flow paths from active flow fronts, new vents and

overflow locations were simulated, so as to inform about likely future lava flow directions. These maps were useful to assess the related hazard and provided situation awareness to stakeholders.

On Mount Etna, Italy, the Istituto Nazionale di Geofisica e Vulcanologia (INGV) at Etna Observatory (EO) is using an operational tool which combines satellite-derived discharge rate estimates and the MAGFLOW numerical code (Vicari et al. 2011, Ganci et al. 2012). The EO provides the simulation outputs to the Italian Civil Protection within the framework of an

operational agreement aimed at minimizing the impact of lava flows.

At Piton de la Fournaise (La Reunion, France), the local Observatoire Volcanologique du Piton de la Fournaise (OVPF) is tackling the hazard related to the frequent effusive eruptions by combining the processing of satellite data with numerical lava flow modeling (Harris et al. 2017, 2019, Peltier et al., 2022). The OVPF has promoted an effective collaboration between scientists affiliated to a multinational array of institutes and is able to quickly issue hazard maps based on

DOWNFLOW (Favalli et al., 2005) and PyFLOWGO (Chevrel et al., 2018) within a few hours after the onset of an effusive eruption (Peltier et al., 2020).

## 1.2 Lava flow hazard modelling in Iceland

Lava flow simulation during an eruption in Iceland was first done during the 2010 Fimmvörðuháls eruption with VORIS, which is an automatic GIS-based system for volcanic hazard assessment (Felpeto et al., 2007, Thorkelsson et al., 2012). The

simulation was made to describe a potential scenario, with assumed eruption location and runout length as input parameters. This scenario was not updated as the eruption progressed. Prior to the onset of the 2014-2015 Holuhraun eruption, VORIS was ran as part of the VOLCANBOX package (https://volcanbox.wordpress.com/) within the VeTools project (http://www.evevolcanoearlywarning.eu/vetools-objectives/) and a new Python-based stochastic model, MrLavaLoba started being run (de' Michieli Vitturi and Tarquini, 2018; Tarquini et al., 2019). During the unrest phase prior the Holuhraun

eruption, both VORIS and MrLavaLoba were run regularly and compared to each other. MrLavaLoba continued to be developed and improved throughout the eruption (Tarquini et al., 2019). After this eruption, both VORIS and MrLavaLoba





were used for Icelandic hazard assessment projects (https://skemman.is/handle/1946/24831, https://skemman.is/handle/1946/30779, https://www.vedur.is/media/vedurstofan-utgafa-2020/VI_2020_011_en.pdf).

## 2 Geological setting and eruptive history

Reykjanes Peninsula is an oblique spreading zone, characterized by eruptive fissures, open fissures and N-S striking strike-slip faults that are associated with the Mid-Atlantic plate boundary (e.g., Klein et al., 1977; Gee, 1998; Clifton and Kattenhorn, 2006; Einarsson et al., 2020, Sæmundsson et al., 2020). The eruptive centers have been divided into 4 – 6 volcanic systems (Fig. 1c), based on high-temperature geothermal areas, magnetic anomalies, eruptive centers, and geochemistry and are from East to West named: Hengill, Brennisteinfjöll, Krýsuvík, Fagradalsfjall, Svartsengi and

Reykjanes (e.g., Jakobsson et al., 1978; Einarsson and Saemundsson, 1987; Einarsson et al., 2020, Sæmundsson et al., 2020).

The last four thousand years, volcanic activity on the Reykjanes Peninsula has been episodic, with several eruptions occurring in multiple volcanic systems over several hundred years followed by ~800–1000 years of quiescence. During this time Reykjanes, Svartsengi, Krýsuvík, Brennisteinsfjöll and Hengill volcanic systems were active, while the Fagradalsfjall

volcanic system remained inactive (Sæmundsson et al., 2020). The last eruptive period ended in 1240 CE (Sæmundsson et al., 2020). Basaltic subaerial volcanic activity has dominated the Reykjanes Peninsula since the termination of the last glaciation, estimated at around 12,000 – 15,000 years ago (e.g., Jakobsson et al., 1978; Saemundsson et al., 2010). The axial centers of the volcanic systems are dominated by eruption fissures, while shield volcanos lie on the periphery of each swarm (Jakobsson et al., 1978). The fissure eruptions were presumably short-lived, high effusion rate eruptions, while the shields

are believed to be long-lived monogenetic eruptions that dominated the early postglacial times (Rossi, 1996, Jakobsson et al., 1978). During interglacial periods volcanic eruptions formed widespread glaciovolcanic edifices on the peninsula ranging from small mounds, tindars, flat-topped tuyas to multiple, polygenetic complexes of intergrown tindars and tuyas (Jones, 1969, Saemundsson et al., 2010; Pedersen and Grosse, 2014). Mt. Fagradalsfjall and close surroundings is a complex of intergrown tuyas, tindars and mounds of different ages creating a topographic diverse area with mountains ranging from 100-

350 m elevation cut by nested enclosed valleys ranging from 50–215 m elevation. Around this glaciovolcanic complex there are postglacial lava fields gently dipping away from the complex in all directions.

### 2.1 Fagradalsfjall unrest and eruption

Prior to the eruption unrest was detected at multiple volcanic systems (Svartsengi, Reykjanes and Krýsuvík) along the

Reykjanes Peninsula revealed by intense seismicity that started in December 2019 and ground deformation revealing inflation and deflation episodes starting in January 2020 (Floventz et al., 2022, Geirsson et al., 2021, Barsotti et al., review).



On February 24, 2021 an intense earthquake swarm began with a magnitude 5.6 located 2-3 km NE of Fagradalsfjall marked the start of a dike intrusion. The dike continued to lengthen and widen during the next 3 weeks before it ruptured East of Fagradalsfjall.

The eruption began on March 19 between 20:30 to 20:50 UTC in the Geldingadalir valley when a 180 m long fissure opened. The fissure quickly concentrated into a few vents, which after 2 weeks had concentrated on two neighboring vents. The lava started infilling the valley with a fairly low time-average discharge rate (TADR) ranging from 1–8 m$^3$/s (Pedersen et al., in press). By April 5 a new phase of eruptive activity started as two new fissures opened 800 m northeast of the first fissures.

Another fissure opened at midnight on April 7, one on April 10 and then two new fissures opened on April 13. Each fissure concentrated into 1–2 circular vents, which over the following 10 days became inactive, except for southern vent that developed from the April 13 fissures. By April 27 only one vent, which opened on April 13, was active, and remained the source of lava effusion throughout the rest of the eruption. During this vent migration phase the TADR ranged from 5 to 8 m$^3$/s and the lava started to flow into the valleys of Meradalir (April 5) and Syðri-Meradalur (April 14). From April 27 to

June 28 the TADR increased from 9 m$^3$/s to a maximum of 13 m$^3$/s and with this increased effusion rate the lava migrated to its maximum extend 3.3 km from the active vent through lava transport systems of connected channels, lava ponds and tubes (Pedersen et al., in press). The lava "filled and spilled" to Nátthagi valley through Syðri-Meradalur (May 22) and through southern Geldingadalir (June 13). From June 28 to September 2 the lava effusion from the vent changed from being continuous to episodic with intense lava emplacement (ca. 12–24 hours) followed by inactive periods of similar length.

Despite this change, the recorded TADR in this phase is similar to the previous phase ranging from 9 to 11 m$^3$/s. The episodic activity disrupted the dominating lava transport system causing large overflows in the vent region where an additional 50 m of lava piled up increasing its total maximum thickness to 124 m (Pedersen et al., in press). In the last days of the eruption, from September 2 to September 18, a 9-day-long pause (September 2–11) was followed by a week-long period of activity from September 11 to September 18. Most of the deposition was in Geldingadalir, where a 10–15 m thick

lava pond was established north-northwest of the active crater between September 11 to 15. The pond partly drained through an upwelling zone southward and into Nátthagi (September 15–18). The measured TADR was 12 m$^3$/s for September 9–17 and the final bulk volume of the lava flow-field increased to 150.8 × 10$^6$ m$^3$ covering an area of 4.85 km$^2$ (Pedersen et al., in press).

**3 Data and Methods**

**3.1 Data**

The primary data source for lava flow simulations were pre-eruptive and syn-eruptive digital elevation models (DEMs) that constitute the computational domain for the lava flow simulations.



In addition, in the pre-eruptive phase the lava flow simulations were initialized by using hypothetical scenarios characterized by different fissure lengths and total volumes. The data, publicly available online, were extracted from the Catalogue of

Icelandic Volcanoes which provides three main categories of eruptive scenarios for the volcanic systems considered for the unrest at Reykjanes peninsula (Sigurgeirsson and Einarsson, 2016). Specifically, three total volumes were considered; i.e. <0.1 km$^3$ (small scenario), 0.1-0.5 km$^3$ (medium) and >0.5 km$^3$ (large). The set of simulations were run accordingly with these three volumes. Previous eruptions on the peninsula are also known to have been featured through single vents, short fissures (2 km) and/or long fissures (10 km). Given the uncertainty in the eruption setup during the unrest phase, a plethora

of runs were undertaken to investigate the potential extension of lava flows for a combination of these parameters. Once the eruption started, the available measurements of extruded volume and emitting vent geometry, were used for initializing simulations to produce the short-term and long-term hazard assessment.

### 3.1.1 Pre-eruption DEM

As pre-eruption DEM, we used the 2 m-cell size IslandsDEMv0 (atlas.lmi.is/dem), a seamless mosaic of the ArcticDEM

(Porter et al., 2018) with an improved positional accuracy and reduced amount of data outliers. Based on comparisons with lidar surveys carried out in the vicinity of the Icelandic glaciers (Jóhannesson et al., 2013), the elevation accuracy of the pre-eruption IslandsDEMv0 is better than 0.5 m (https://gatt.lmi.is/geonetwork/srv/eng/catalog.search#/metadata/e6712430-a63c-4ae5-9158-c89d16da6361). The cell size of the computational domain (i.e., the DEM in grid format representing the local topography) has a strong impact on the performance of the MrLavaLoba code. For a given areal extent of the computational

domain, a smaller cell size results in a higher the total number of grid cells, and thus in longer simulation time. Therefore, the IslandsDEMv0 was downsampled to 5 m and 10 m cell size grids depending on the expected extent of simulated scenarios. Large long-term scenarios (volume > 50 Mm$^3$) were simulated on 10 × 10 m spatial resolution, while smaller short-term scenarios (volume < 50 Mm$^3$) were simulated on 5 × 5 m spatial resolution. In a few cases simulating lava flows close to barriers or within a narrow valley setting, the full resolution, 2 × 2 m, of the IslandsDEMv0 was used as computational domain.

### 3.1.2 Syn-eruption photogrammetric surveys

Throughout the eruption, photogrammetric surveys were acquired as a part of the near real-time monitoring of the Fagradalsfjall 2021 eruption. These surveys consisted mainly of aerial photographs and Pléiades stereoimages and by September 30, 2021, 32 syn-eruptive surveys had been carried out. The acquisition and processing of these surveys are described in detail in Pedersen et al. (in press) mainly following the semi-automated workflow of Belart et al. (2019) using the

software MicMac (Pierrot Deseilligny et al., 2011, Rupnik et al., 2017), as well as Agisoft Metashape (version 1.7.3) and Pix4D mapper (version 4.6.4). Each of the surveys were co-registered to the pre-eruption DEM, i.e., the IslandsDEMv0, using the DEM co-registration method of the Nuth and Kääb (2011).





Each survey yielded DEMs ($2 \times 2$ m) and orthomosaics ($0.3 \times 0.3$ m) from which the lava flow outline was obtained. By subtracting the DEMs with a pre-eruption DEM and with the DEM from the previous survey it was possible to obtain thickness

maps ($2 \times 2$ m) and estimate bulk eruption volumes and time-averaged discharge rates (TADR). These data products were generally available 3–6 hours after acquisition.

The thickness maps proved valuable for the lava flow simulation. Not only as comparison to the lava flow simulation, but as the lava field became increasingly complex (after April 27), they were also implemented as a part of the computational domain for the short-term simulation updating the topography to the most current survey.

**3.2 Methods**

3.2.1 Software

MrLavaLoba is a probabilistic lava flow simulation code that was developed by Mattia de' Michieli Vitturi and Simone Tarquini from INGV, Italy starting in 2014. The code was published in 2018 (de' Michieli Vitturi and Tarquini, 2018) and is freely available at the model repository (http://demichie.github.io/MrLavaLoba/) and it has previously been applied to the

following eruptions: Etna 2001, Kilauea 2014–2016 and Holuhraun 2014-15 (de' Michieli Vitturi and Tarquini, 2018; Tarquini et al., 2019). Unpublished tests have also been carried out at Piton de la Fournaise (La Reunion, France) and on the 2014-2015 Fogo eruption (Cape Verde).

In MrLavaLoba the lava emplacement is largely driven by the slope of the topography and by tunable input settings, while lava „parcels" are deposited along the flow path enabling continuous modification of the topography as the lava is deposited.

In this way the code mimics that lava flows constantly create new topography within or on which new lava flows or lobes are deposited. Each chain of parcels making up a flow path is called a "flow" and the number of flows is an input parameter of the code. The code, given the lava volume, provides the final emplacement thickness of a lava flow field.

Beside a computational domain constituted by the pre-emplacement topography in grid format, the code requires to set a

series of input parameters (including vent position, area of the parcels, cumulative volume, parameters that mimic style of emplacement, etc.). Examples of input settings written in a largely commented Python code can be found in the code repository (http://demichie.github.io/MrLavaLoba/).

Additional topographical layers can also easily be included in the model, such as lava thickness maps from syn-eruptive surveys or lava barriers. When running the model, the lava is emplaced stepwise as elliptical parcels. The emplacement is the

process of budding new parcels from the existing ones. The direction of propagation of the flow is determined by the direction of the steepest path ($az_i$) (derived from the emplacement topography) with the addition of a random perturbation ($e_{az}$) and an "inertial factor" which considers the direction of the parent parcel ($az_p$). Once the direction of propagation is determined, the new parcel is added in its final position. The area and thickness of each parcel is then added to the



emplacement topography reflecting the deposition of lava, which constantly changes the pre-emplacement topography.

Another stochastic variable, the "lobe exponent", controls the probability distribution among the existing parcels to bud a new lobe (when this parameter is set to 0, the latest parcel emplaced generates the next lobe and no branching occurs). The number of parcels in each flow, and the number of flows are input parameters in the code. The code proceeds by iteratively setting new parcels on the topography until their total volume equals that prescribed volume for the simulation. Several further tuning options are implemented to mimic different lava transport mechanisms (channelized flow, lava tunnels or

stochastic budding of lava lobes) accounting for a given propensity to lengthening, widening or thickening of the flow field. In addition to the full inundated area, the code allows saving masked grids obtained by considering inundated cells fulfilling a specified threshold value. If this threshold is set to 0.95, the thinnest portion of the final lava deposit representing 5% of the total volume is disregarded from the results. This step is important due to the probabilistic nature of the code, where the thinnest part of the inundated area represents a lower probability for inundation and may change from one simulation to

another given the same input parameters, while the masked area represents an area "more likely" to be inundated. Through iterations of a large number of flows, MrLavaLoba handles the probabilistic aspect of lava emplacement.

During the Fagradalsfjall eruption several new features have been implemented to improve its applicability to the continuously changing conditions. One of the first changes was to add the possibility to have multiple vents (or multiple

fissures) active at the same time and with a prescribed supply probability. Secondly, the code was modified to enable multiple threshold values for a single simulation, in order to filter inundated areas according to different levels of probability of inundation given a set of input parameters. Finally, several code optimizations have been done to accelerate the code, both in the input/output procedures and in the computation of the flow emplacement. With respect to the version available at the beginning of the volcanic crisis, the code now is up to 7 times faster.


### 3.2.2 Implementation

The implementation of MrLavaLoba code depended on the purpose of the simulation and and table 1 provides a general overview of the simulation goals, approaches and time-dependent/varying input parameters, while the full set of input parameters can be found in table A1.

Since MrLavaLoba is a stochastic code, it cannot provide the temporal evolution of the flow field for each run. However, since volume is one of the input parameters, the temporal aspect of lava flow-field evolution can be addressed by simulating various volumes and have input parameter such as number of flows (n_flows) and lobes per flow (min_n_ lobes & max_n_lobes) scalable based on the effusion rate and time. Thus, a higher effusion rate would provide more flows from the vent and longer flows (so higher n_flow number and higher min_n_lobe number), and with time the number of lobes would also increase

(higher min_n_lobe number). In this way, insight to the temporal evolution of the lava field could be addressed either by pre-



defined effusions rates, as used in the pre-eruption simulations, or based on measured TADR during syn-eruptive simulations. How these input parameters were scaled with effusion rate and time changed from the pre-eruptive to the syn-eruptive simulations. Both because there was more than one order of magnitude difference in effusion rate between the worst-case scenarios simulated in the pre-eruption simulations (300 m³/s) and observed TADR during the eruption (mean TADR for the

eruption was 9.5 m³/s), but also because it was much harder to evaluate the results from the pre-eruptive scenarios compared to the syn-eruptive scenarios, where lava flow simulations could be tuned to the observed lava flow thickness maps.

## 4 Results

Different lava simulation strategies were implemented during the unrest and eruption depending on purpose of simulation (Table 1). Overall, the purposes were three-fold; (a) Pre-eruption simulation to investigate potential infrastructure in

immediate danger for lava flow inundation based on location of deformation signal (b) Simulations addressing areas of short-term danger for lava flow inundation (two weeks) and (c) simulations addressing areas of danger for lava flow inundation in long-term (months to years).

### 4.1 Pre-eruption simulations

During the unrest pre-eruption lava flow simulations were initiated after InSAR data from February 23 to March 1, 2021,

revealed crustal deformation consistent with a 9 km long dike intrusion causing intense seismicity (Geirsson et al., 2021). The location of the seismicity was both associated with the dike intrusion, but also located on neighboring faults which were triggered by stress changes in the crust and not related to the intrusion of magma directly. As a result, seismicity alone wasn´t specific enough to indicate where the dike that eventually erupted were migrating, and therefore a combination of seismic observations, deformation observations (cGPS and InSAR), stress modelling, and deformation modeling (Geirsson et al., 2021)

gave the best indication of potential fissure openings. Based on this information 12 different dike openings of 2-10 km length were chosen for pre-eruptive lava flow simulations (Fig. 2). These lengths were chosen with respect to length of visible eruptive fissures on the Reykjanes peninsula based on data from Jónsson (1978). It is considered very unlikely that an eruptive fissure of 10 km length will erupt on the western part of the Reykjanes peninsula but with respect to the worst-case scenarios a few lava flow simulations were run using this fissure length.

Two different strategies were implemented in this unrest phase; one short-term worst-case scenario addressing areas likely to be inundated within a few hours from eruption start relevant to emergency response planning and one for longer-term scenarios providing insight to areas likely to be inundated within weeks to months functional for identification of infrastructure at risk.

### 4.1.1 Short-term worst-case scenario





The worst-case scenario was defined as a fissure with an effusion rate of 300 m³/s, where the number of flows, n_flows = 300 and min_n_lobes were multiplied with 3.33 * per minute to mimic the lengthening of the flows. This multiplication factor was estimated to be rather high based on the run-out distance from the vents after each time interval but was preferred rather than being too conservative.


Using this approach, it was possible to evaluate areas of likely inundation hours after opening of a worst-case scenario for the 12 defined dike openings (see example Fig. 2b).

Based on the selected fissure openings and model set-up the results suggested that no inhabited areas were in immediate danger the first hours, and infrastructure was only in danger the first hours if the dike continued propagating south cross-

cutting a nearby highway (Fig. 2a). However, an obvious caveat with this strategy was that the only way to validate the chosen parameter space was based on run-out distance and thickness of final deposit.

### 4.1.2 Long (er)-term scenario

Based on knowledge on lava volumes on the Reykjanes peninsula long term scenarios have been classified in three

categories, small (<0.1 km³), medium (0.1-0.5 km³) and large (>0.5 km³) ([www.icelandicvolcanoes.is](http://www.icelandicvolcanoes.is)). During the pre-eruption phase the small and medium scenarios were simulated to evaluate potential endangered infrastructures, since no large eruption scenario is known from the western part of the Reykjanes peninsula. Two different scenarios were run: small eruption scenario using the historical lava Illahraun (Volume= 0.02 km³) as a reference, and moderate eruption scenario using the historical Arnarseturshraun (Volume= 0.3 km³) as a reference. These scenarios were tuned to simulate lava length

from 1-12 km with 5 km as the most likely result and lava thickness from 1-30 m with 10 m thickness as the most likely result. An example of the moderate scenario is shown in Fig. 2b.

### 4.2 Syn-eruptive simulations: short-term hazard assessments

During the eruption the complexity and demands of the short-term runs increased. Here we describe results from three

different approaches applied during the crisis to address the evolution of eruptive activity and the challenges they posed:

I.   First phase of the eruption: Geldingadalir (March 19-April 5)

II.  Second phase: the vent migration phase (April 5-April 27)

III. Phase three to five: Fill and spill of a highly compound lava flow field (April 27-September 18)



### 4.2.1 First phase (April 5-27)

After the eruption had started, we initiated the first runs of MrLavaLoba March 19 around 22:00 UTC (1.5 hours after the eruption started) using preliminary vent coordinates provided by the Civil Protection obtained on a helicopter flight. However, the precise location and length of the fissure was first acquired the following morning when the first aerial images during daylight had been georeferenced giving precise location and length of the fissure.

The main purpose of these runs was to evaluate how the lava would infill the Geldingadalir valleys and when it would spill

into Syðri-Meradalur East of Geldingadalir, inundating the path for hikers visiting the eruption (Fig. 1, 3). We remark here that MrLavaLoba does not provide a temporal evolution of the lava flow-field, but by simulating different volumes and assuming a range of TADR, constrains on the timing of the spill into Syðri-Meradalur and of the hiking paths inundation can be inferred.

During this phase we used a stationary 180 m fissure erupting equally along the fissure segment, despite the fact that the 180 m long fissure quickly concentrated to a few vents and within 14 days, only two vents were active in the northern end of the fissure. Mimicking the vent concentration from a fissure into a few points within a single simulation would have required a major change in the code, or otherwise would have required to develop a complex, time-consuming, step-wise simulation strategy that was impossible to fit in such "emergency-mode" responding timeline.


The lava simulations were qualitatively evaluated by comparing the thickness maps obtained from photogrammetric surveys with the thickness maps lava simulations. However, for the first hours of the eruption (<12 hr) the only documentation was from a few very oblique photographs. Fig. 3 reveals that the smallest run (V= 0.018Mm$^3$) show northern and southern lobes agreeing with the photographs documenting the extent of the lava at midnight March 19, ca. 3-4 hr after eruption start.

However, for the simulations with a volume between 0.2 Mm$^3$ to 3Mm$^3$, the lava simulations overestimate the extent of the southern extent of the lava field, whilst underestimating the lava thickness of the northern lobe, which can be explained with the closing of the vents to the South of the fissure. However, for the simulations with volume between 3-7 Mm$^3$ the results agree fairly well with the observations suggesting that the lava at this point was so confined by the Geldingadalir valley that the change in vent geometry had little effect on the lava inundation area.


The simulations predicted potential exit from Geldingadalir valley into Syðri-Meradalur valley from 7 – 10 Mm$^3$, with slightly different volumes from run to run due to the stochastic nature of the MrLavaLoba code. The TADR estimates obtained from photogrammetry in this phase ranged from 1–8 m$^3$/s with a mean of $4.9 \pm 0.1$ m$^3$/s. In order to provide potential timing of when Geldingadalir would fill and spill into the Syðri-Meradalur valley we used a maximum effusion rate

of 10 m$^3$/s to provide a minimum time for when the valley potentially would spill. This gave a minimum time of 8-12 days after eruption start (so March 27 -31) before exit from Geldingadalir. For 8 m$^3$/s it would be 10-14 days (March 29 -April 2)





and for the 5 m$^3$/s it would be 16-24 days (April 4-April 12). The lava eventually spilled out from Geldingadalir valley on April 14, but by April 5 new vents had opened North of Geldingadalir, draining the lava supply from the initial vent. The measured total TADR for all vents between April 5-18 was between 5- 8 m$^3$/s, however the majority of the lava at this point

were deposited in Meradalir and the plateau NE of the Geldingadalir vent. By calculating the lava volume within Geldingadalir by April 18 we get a volume of 10.8 Mm$^3$, whilst the total erupted volume was 16.1 Mm$^3$ (first photogrammetric survey after lava exited Geldingadalir 4 days earlier). The 10.8 Mm$^3$ is in the upper end of the predicted volume for lava exiting Geldingadalir of the simulations, but still within reasonable agreement.

**4.2.2 Second phase: Vent migration**

In phase 2 (April 5 to April 27) the active vent migrated (Fig. 1). Multiple eruption fissures opened, starting on April 5, when two new fissures opened 800 m northeast of the first fissure. Another fissure opened at midnight on April 7, another one on April 10 and then on April 13 two new fissures opened. Each fissure concentrated into 1–2 circular vents which, over the following 10 days, became inactive, except for the southern vents that developed from the April 13 fissures. All of these

fissures had variable effusion rates.

This change in eruption activity provided new challenges to the lava flow modelling, which are illustrated in Fig. 4. The first challenge was that the topography drastically changed with the new vent openings. From lavas being strongly constrained within the Geldingadalir valley, the lava was after April 5 issued from the plateau NE of Geldingadalir and channelized into

narrow gullies before spreading out like a fan within the Meradalir valley (Fig 4). The new vents had sufficient spacing that the activity in Geldingadalir and at fissure 2 could be simulated in two different runs (Fig.4, V< 0.4 Mm$^3$). However, to capture the channelizing into the narrow valleys it was necessary to increase the resolution of the computational domain DEM (from 5 m to 2 m cellsize) and change the lobe exponent parameter from 0.07 to 0.03 (thus increasing the probability of new lobes to be generated by younger lobes). Lower DEM resolution and higher lobe exponent caused over spilling from

one valley to another earlier than observed in the eruption.

The second challenge was that after fissure 3 opened on April 7, it was clear that the lava flows from the active vents were influencing each other and it was therefore necessary to simulate multiple vents that emitted variable percentages of the total lava volume. The MrLavaLoba code was then modified to allow this configuration, and from this stage and onwards it was

possible to simulate multiple vents simultaneously. However, there was very little available information on the variable percentages of the total lava volume each vent emitted. Qualitative estimates were made based on webcams and direct observations in the field. As the lava flow field emplacement progressed, it became evident that it was necessary to include the most recent lava thickness maps from the photogrammetric surveys. Thereby computational domain was updated, and





new simulations could be performed on the most updated topography. However, by doing so another problem arose; namely

what we called the "restart problem". After including updated topography, the code would start simulating a new eruption, where lava parcels would be initiated from the vents, meaning that it would require a given number of parcels before the edges of the lava field would be activated again creating a delay in the areal expansion of the simulated lava field compared to the real flow-field.

Figure 4 show two runs with two different vent configurations; one with all vents being active and one performed after the

two northernmost vents had shut down (Fig. 4, lowermost panel). These simulations did show how the lava would expand into the neighboring valleys: Syðri-Meradalur and Meradalir. However due to restart problem, some of the results underestimated the expansion of the lava field into Meradalir, whilst other areas (e.g., Geldingadalir and the plateau NE of Geldingadalir) were overestimated, probably due to incorrect ratios of emitted volume between active vents.

**4.2.3 Third to fifth phase: Fill and spill of highly compound lava flow field**

After April 27 the vent activity stabilized to one location (Fig. 1, Vent 5). The mean TADR increased from 6 $m^3$/s to 11 $m^3$/s. The lava flow-field expanded into neighboring valleys such as Nátthagi and further into Meradalir in a "fill and spill" process. There was great interest in simulations that forecasted when and how the lava might overflow from one valley to another e.g., from Geldingadalir into Nátthagi (cross cutting a popular hiking path) or when the lava would exit from the

Meradalir valley (inundating a dirt road), or when it would exit Nátthagi (threatening a highway and critical communication cables, as well as approaching the sea). However, due to the restart problem (see 4.2.2) and because the lava discharge into different valleys was highly variable switching from one valley to another in an unpredictable manner (Pedersen et al., in press), it was therefore decided to address the short-term hazard of lava exiting from a valley with worst-case scenarios. These scenarios were presented at bi-weekly stakeholder meetings, where the aim was to (i) identify hazardous areas; (ii)

create awareness of potential upcoming inundation of hiking paths, roads and installed infrastructure, and (iii) suggest where to close areas closed for public access (e.g., closing of Nátthagikríka in September 2021).

In these simulated scenarios, the TADR estimate based on the most recent photogrammetric survey was used to calculate lava volumes equal to set time periods of 3, 7, and 14 days. These volumes were then released at critical lava front in order

to evaluate if each given volume was sufficient to overflow the valley. If a 3-day scenario would spill out of the valley, then 6 hr, 12 hr and 24 hr scenarios would be modelled as well. The critical locations were selected by the modeler qualitatively based on knowledge of hiking paths and infrastructures.

In these runs, the input parameter "number of flows" was doubled (from 80 to 160, Table 1), both because of the increase of the TADR, and because we found that having more than 100 flows results in a reduced uncertainty in the simulation outputs

(Fig. 7 in de' Michieli Vitturi and Tarquini, 2018).





An example of how these worst-case scenarios were presented at the meetings with the stakeholders is provided in fig. 5. To simplify the maps, we decided only to show the lava inundation area and not the simulated lava thickness maps. In this way, the results for the volumes to 3, 7 and 14 days could be displayed in one map. The main map (Fig.5a), show simulations

from vent 5 and was considered the most likely scenario, while the four smaller panels show the same volumes released at the defined critical locations (Fig.5b-e). As it can be seen by the provided example from September 9, 2021, both the Meradalir, Geldingadalir and Nátthagi valleys could overflow, given the lava was transported to the critical points that were used as simulated vents.  The main weakness of this approach is that the hypothetical outbreaks areas arbitrarily selected on the basis of the available knowledge and are subject to a large uncertainty. Furthermore, re-tuning of the code to simulate

lava poured from lava front edges rather than the actual vent is a more difficult task, since most of these hypothetical outbreaks did not happen.

A similar approach was used to test barriers that were build or planned to be built during the eruption. An example can be found in fig. 6, where the overflow of Geldingadalir was simulated with and without barriers based on photogrammetric data

from a survey on June 11, 2021. All scenarios show that with 1 Mm$^3$ volume lava Geldingadalir will overflow into Nátthagi. For no barriers it will also spill west into Nátthagikríka, but with barriers it seemed plausible to stall the west-ward migration, at least for small volumes. Once again, these were worst-case scenarios, because (i) they require that the given volume of lava is transported to the simulated vent, and (ii) assume that the transport systems near the barrier is not efficient and promotes the lava piling up near the barrier. However, if an efficient lava transport system develops, a little amount of

lava would pile up close to the barriers, meaning that they would last longer.

### 4.3 Long-term runs

The long-term scenarios were planned to be released to end-users (managers of critical infrastructure, municipalities, civil protection authorities) in September 2021 and were requested from stakeholders for longer term planning. However, the

eruption came to a halt on September 18, and these scenarios were not officially released but just presented at meetings with stakeholders. Long-term scenarios were simulated by considering erupted lava volumes ranging from 250 Mm$^3$ to 5000 Mm$^3$. If the mean TADR is assumed to be always equal to the mean TADR for the entire eruption (9.5 m$^3$/s, Pedersen et al., in press), it turns out that the simulated long-term scenarios cover time frames of half a year to decades.

To simplify these large simulations only one vent was considered active (vent no. 5 in Fig. 1) throughout the simulation, which had been the case since April 27, 2021. Furthermore, each scenario was obtained as a single run (with a single tuning) and not as a series of runs with an iterative process of tuning the model step by step. We tuned the long-term runs against the



lava thickness maps obtained in June having volumes between 53-80 Mm³ and preferred to overestimate rather than underestimate lava deposits. Fig. 7 shows the thickness maps obtained by the airborne photogrammetry and the simulation

results for 80 Mm³. As can be seen this model set-up does not include lava on the northeast plateau north of vent 5, which is due to the fact that the long-term simulation is based on a single active vent (Fig.7, top row). Generally, the fit is acceptable, although the thickening close to the vent was underestimated and the thickening in the Meradalir valley was overestimated.

These long-term models were specifically intended for stakeholders with no experience with lava flow simulation. The new

challenge was therefore how to communicate the uncertainty in our results to non-experts (Pallister et al., 2019). We decided to create maps where lava inundation would be divided into three categories: "very likely", "likely" and "less likely" (Fig.7, bottom row). Thus, like the worst-case scenario maps produced in phase 3-5 we would not display the thickness maps, but simply the lava inundation area, since this is of main interest to stakeholders.

The code of MrLavaLoba is not designed to communicate uncertainty of the results, and it was therefore necessary to come up with a strategy to define these categories. The full extent of the lava simulation was decided to be categorised as "likely". This result would be based on the input parameters derived from the tuning, but the extent of the thinnest lava deposits would vary from simulation to simulation due to the stochastic behavior of the code. de' Michieli Vitturi and Tarquini, (2018) noted that by using a 95% mask showing the thickest 95% of the deposit (i.e., disregarding the 5% of volume given

by the thinnest part of the deposit) the results from different simulation would converge to a given coverage. Thus, by using the 95% mask the results are filtered removing the places least likely to be inundate and it was therefore decided to use the 95 % mask as the "very likely" category.

However, we also wanted to communicate the uncertainty related to the model tuning. Especially because the tuning

parameters were derived when the lava flow field were highly constrained by the surrounding topography (Fagradalsfjalls mountains and valleys) whilst the large scenarios would inundate the low sloping areas outside the Fagradalsfjalls area, where the lava flow field could spread more freely over flat lying areas. However, at this point we did not have a quantitative approach for addressing this issue, and instead, it was decided to change the tuning of the "lobe exponent" which is crucial parameter for the lava flow spreading. A lower lobe exponent would promote lower flow thickness and a longer run out

length. Choosing a lower lobe-exponent (0.01 instead of 0.02) that largely overestimate the inundated area of our tuning data set (Lava thickness maps from June) were therefore used as a route to communicate scenarios that, based on our tuning data set, were "less likely", but could not be excluded due to the very changeable topography in the area (Fig.7, top row).

One example of the produced long-term maps can be found in Fig. 8. The main take home message from these long-term

simulations is that none of the runs reach Grindavik nor the Svartsengi powerplant, which are two of the main inhabited and infrastructure areas closest to the eruption site. However, it is important to underline that these long-term runs are quite



uncertain, because as the scenario grows in volume so does the uncertainty. In addition to the tuning being undertaken while the lava was highly controlled by topography, the following additional challenge was encountered. The style of eruptive activity varied between continuous to fountaining to episodic activity (12-24 hours of quiescence). These changes impacted

the efficiency of the lava transport mechanisms (Pedersen et al., in press). When the tuning was performed, the activity was fountaining and had transport systems that enabled lava emplacement from the edges of the lava flow field about 3.3 km from the vent. In July, instead, the activity was episodic, and resulted in a shortened lava transport systems with lava preferentially stacked close to the vents. Another point of concern is that the current version of MrLavaLoba does not include vent processes such as cone build-up. This was important because the real vent built up faster than predicted in the

simulation, meaning that the lava flows might had inundated Fagradalsfjall faster than predicted in these models.

## 5 Discussion

How the MrLavaLoba code was implemented and developed during the pre-eruptive and syn-eruptive stage of the Fagradalsfjall 2021 crisis has been described in the results. The flexibility of the code made it possible to implement it in different way for different purposes, which is unique amongst the existing stochastic codes. The MrLavaLoba code is freely

available, easy to run in Python, and computationally fast (especially after the optimizations carried on during this work). It can thus be used to tackle large volume scenarios (Table 2). Unlike other stochastic codes it includes lava volume and final lava extent, it produces lava thickness layers and it models ongoing topographic changes during the simulation. This was very important during the Fagradalsfjall 2021 crisis, because the hazards related to the fill and spill of nested valley systems could not have been addressed by codes that do not include lava volume. It was easy to implement topographic changes,

including the syn-eruptive differential DEMs of the lava flow thickness and lava barriers, which was key for testing suggested and built lava barriers.

However, there were also drawbacks relying on the MrLavaLoba code; including the fact that it is a stochastic code and not a physical code, such that it does not provide temporal evolution of the lava field during each run, nor does it directly provide velocity estimates of the lava emplacement (Table 2). Input parameters must be tuned for specific eruptive scenario and

location. The tuning will be different for different volcanic systems, different topographic conditions and different sizes of scenarios. It is possible to mimic different lava emplacement processes (e.g., lava channels or tubes), but this has to be tuned as well and ideally all of this tuning has to be completed before a volcanic unrest. Furthermore, it became a concern for the long-term scenarios, that the code does not include vent-processes, which lead to an underestimation of the lava thickness close to the vent. This impacted the forecast for when Geldingadalir valley would be filled and thus when lava could migrate

westward over the Fagradalsfjall plateau towards critical infrastructure. However, to our knowledge no lava simulation code includes vent-processes, so this issue would also have applied to all other lava flow simulation codes. Probably, this is mainly an issue for eruptions in flat terrain or within nested valleys, and not on steep slopes that will dictate the flow direction and deposition independent of localized changes at the vent.





The following discussion will focus on the lessons learned so far; the improvements of the code, the implemented modelling approaches and the dissemination of results for hazard communication purposes. We will discuss the current caveats and how the code, approaches and dissemination strategies could be improved for the next volcanic crisis.

**5.1 MrLavaLoba code**

Three main issues with the code were identified and addressed during the Fagradalsfjall 2021 eruption, namely (a) implementation of multiple vents, (b) implementation of multiple masks and (c) scaling issues with changes of resolution of the computational domain. Here we describe how these issues were addressed.

During the vent migration phase (Phase 2) multiple vents were active at the same time, and it was therefore needed to
simulate multiple vents in the same run, allowing lava emplacement from multiple source locations to influence each other. The model capability has been described in section 3.2.1 and was used from the beginning of April and onwards. This model option requires, in addition to the coordinates of all active vents, their relative supply probability, which is a ratio defining how the supply rate is divided among the active vents. Two issues were found with this implementation. First of all, retrieving information on supply probability was difficult based on observation and thus the modeller was forced to make
very rough estimates of this ratio. This challenge reflects the interplay between the limitations of lava flow monitoring and the issues for the person modelling. Secondly, in the eruption, different vents turned on and off at different times, but this temporal source variation is unlikely able to be implemented in MrLavaLoba because there is no „time" in the simulations. In practice, the modeler began a new lava simulation when the number of active vents changed.

In the beginning of September, issues about communicating uncertainty came to the forefront. The model was upgraded to save multiple mask output levels, which has been described in section 3.2.1. This feature was primarily used after the eruption ended to show the full extent (100 % mask) of the lava flow simulation, the 95% mask and e.g., the 68% mask on a map indicating the likelihood of inundation. The lowest mask threshold indicates the thickest lava deposits and thus the areas, that are most likely to be inundated, and the three masks were therefore used to communicate the uncertainty related to
the fact that MrLavaLoba is a probabilistic model and is specific to a given set of input parameters. In the future the quality of these mask thresholds should be tested routinely by repeating each simulation scenario to ensure that the masked results of repeated runs are consistent. Such a consistency indicates that the number of flows in a scenario is adequate for the given thresholds to indicate likelihood (de' Michieli Vitturi and Tarquini 2018).



Some of the input parameters are sensitive to the spatial resolution of the computational domain and thus need to be re-tuned
if the input DEM resolution is changed (for example if using a coarser DEM over a larger domain for long-term scenarios). It
is time-consuming to tune the input parameters, and this can be impractical while a volcanic crisis is ongoing.

In the specific case of no branching (lobe_exponent=0), we observe that the maximum runout of a simulated lava flow is
proportional to the major axis of the elliptical parcels and the number of lobes in each lobe chain. Thus, when the DEM

resolution is changed, and the lobe area is scaled proportionally to optimize the computational cost, the number of lobes in
each chain also must be changed in order to attain a similar runout. In this case, if the lobe area changes by a factor G, the
number of lobes should change by a factor 1/sqrt(G). When lobe_exponent>0, the definition of the scaling factor is less
obvious. In fact, the larger the number of lobes, the larger is the occurrence of branchings, and a tuning is needed to
reproduce the same runout.

**5.2 Modelling approach: pros and cons**

Both in the pre-eruptive and syn-eruptive phases of the Fagradalsfjall 2021 volcanic crisis the modelling approaches were
divided into two categories; 1) a short-term scenarios addressing hazards on a time scale of hours to days and 2) long-term
scenarios focusing on hazards relevant on timescales from months to years.

These scenarios were evaluated qualitatively (table 1), but ideally a quantitative approach should have been implemented
beforehand. Thus, for the scenarios where it was possible to compare simulated lava thickness maps with lava thickness
obtained by photogrammetry it would have been viable to quantitatively compare the two rasters with respect to the accuracy
of (a) estimated lava inundation area and (b) lava thickness.

For future eruptions it will be possible to use the Python script *union_diff.py* available in the model repository

(http://demichie.github.io/MrLavaLoba/). It compares raster files defined on the same grid, meaning that they have the same
spatial resolution and spatial extent and produces different outputs.

First, it computes the Jaccard similarity index, defined as the area of the intersection of two deposits divided by the size of
their union. This index, as discussed in de' Michieli Vitturi and Tarquini (2018), can be used to assess the convergence of the
masked outputs when the number of flows is increased. The script also computes the average thickness difference between

two rasters, and thus it can be used not only to compare two outputs of the model, but also simulated thickness maps with
observed lava thickness.

Another caveat with our strategy was that there was no existing procedure for uncertainty testing of input parameters, nor did
we have a template of how we wished to communicate the uncertainties of the model results. In order to evaluate the

uncertainty of the input parameters themselves, statistical tools such as Dakota (Adams et al., 2021), specifically designed to
perform sensitivity analysis and uncertainty quantification with existing numerical codes, could be used. This would allow to
assess the most critical input parameters and develop map layers to communicate the uncertainty of the parameter space.



Finally, updating the computational domain through incorporation of ancillary data such as syn-eruptive DEMs or lava
thickness maps in lava simulations have been thought of as a route to improve lava flow simulations bypassing caveats of
lava simulation codes to reproduce the lava flow-fields more consistently (Harris et al., 2016, Tarquini et al., 2018).
However, as we discovered, updating the topographic domain created a new problem: namely the restart problem described
in 4.2.2, where the real eruption was continuously transporting lava to the edges of the flow-field while the re-initialized
simulation experienced a delay in the lava field expansion as the simulation required a „spin up time" to get lava advancing
at the edges. We have learned that discretizing lava simulations into sequential separate simulations with updated topography
will require a strategy to deal with this "spin up time". This issue is not specific to the MrLavaLoba code but will likely
apply to all lava simulation codes that attempt to sequentially introduce new starting conditions to implement time evolution.

We propose some potential ways forward to solve this:

1) integrate lava transport system

2) Find a way of having a „spin up" that will re-establish activity at the flow field edges.

3) Implement additional sources at active lava margins as we did for the worst-case scenario runs described in 4.2.3.

**5.3 Dissemination: Communication & Hazard maps**

Based on the bi-weekly meetings with stakeholders it became clear that efficient lava flow simulations and dissemination of
results relied on understanding the needs of the stakeholders: (a) what information did they want or need (e.g., inundated
area, time frames of processes), (b) who are they and what is their level of engagement with the material (e.g., experts, civil
defence/responders, decision makers or the general public) and (c) what type of product/map would be best understood given
what was known about (a) and (b). For the eruption at Fagradalsfjall 2021 maps were developed as the crisis continued and
as feedback was received from the stakeholders in regular meetings. The main products were short-term more-likely
scenarios, short-term worst-case scenarios (Fig.5) and long-term scenarios (Fig.8).

In our case of a long-lasting volcanic crisis, the information had to be communicated in brief and efficient meetings with
agencies responsible for the operations, as well as during some ad-hoc meetings with key stakeholders, dedicated to discuss
the most important results shown in the lava flow maps. It turned out the most efficient way to communicate results were (1)
simple maps that showed potential lava inundation areas, and not lava thicknesses, (2) to have uncertainty related to the
model results included on the maps, and (3) that all necessary information to understand the results have to be on the map (as
opposed to in a figure caption, for example), so that key- information (e.g., name of lava simulation model, key input





parameters, time-frame and main assumptions) could not be inadvertently separated from the shared results and therefore the
final product would be self-explaining.

As described above (section 5.1 and 5.2) including the uncertainty of the lava simulation results, required changes to the
simulation code, and in the modelling approach, which, to begin with was requiring extra time. However, it also became
clear that if we had efficient ways to include uncertainty in the simulation results, the modeler could (i) avoid time-
consuming fine tuning of input parameters, (ii) save time in the map production and (iii) be much more efficient in
communicating the results in a consistent way.

For the next volcanic crisis, we aim to have pre-made map templates for short-term scenarios, short-term worst-case
scenarios and the long-term scenarios. These should be developed collaboratively with hazard and cartography experts to
help the lava flow modeler find a suitable balance between essential information and simplicity. Secondly, an explanation of
the lava hazard modelling approach should be prepared in multiple languages to accompany the maps in the stakeholder
meetings. Finally, it would be helpful to discriminate between design of map products delivered to 1) scientific community,
2) stakeholders and 3) the general public and potentially set-up some a priori agreements about what type of results should
be disseminated to each group and consider appropriate the platform for such information (websites, wms, social media).

**6 Conclusion**

On March 19 to September 18, 2021, the first eruption in ca. 800 years took place on the Reykjanes Peninsula, Iceland.
Being located in the backyard of the capital Reykjavík and the international airport, this effusive eruption was the most
visited eruption in Iceland to date and needed intense monitoring and thorough lava hazard assessment. Furthermore, it
became a test case for the monitoring and hazard assessment for future eruptions on the Peninsula that can have a greater
societal impact issuing lava into inhabited areas or inundate essential infrastructure.

In this study we documented how various lava simulation strategies using the stochastic code MrLavaLoba was a useful and
a flexible tool to evaluate hazards during 6-month long effusive event. Different lava simulation strategies were deployed
during the unrest and eruption depending on purpose of simulation. Overall, the purposes were three-fold; (a) pre-eruption
simulation to investigate potential infrastructure in immediate danger for lava flow inundation based on location of
deformation signal (b) Simulations addressing areas of short-term danger for lava flow inundation and worst-case scenarios
(Two weeks) and (c) simulations addressing areas of danger for lava flow inundation in long-term (Months to years).

The functionalities of the code made it flexible and possible to implement it for multiple purposes. Unique to other stochastic
codes, MrLavaLoba includes lava volume, final lava extent, it produces lava thickness layers and it changes topography





during the simulation. This was very important during the Fagradalsfjall 2021 crisis because the hazards related to the nested valley systems that were filling and spilling from one valley to another could not have been addressed by codes not including lava volume. Furthermore, it is very easy to implement topographic changes, e.g., by implementing lava flow thickness maps obtained during the eruption or lava barriers, which were key for testing suggested and build lava barriers.

Tuning of the input parameters needed for the MrLavaLoba code was at times time-consuming, especially during changeable eruptive activity or changes in terrain and ideally tuning of a greater number of eruptive scenarios in different terrain has to be prepared beforehand. A couple of other issues discovered during the crisis was the restart problem when updating the topographic computational domain. After updating the topography and restarting the lava flow simulation a delay in the lava field expansion were observed in the simulated lava field while the real eruption continuously were transporting lava to the

edges. Another issue discussed was the scaling issues caused by changing spatial resolution of the computational domain.

During the crisis the code was updated to increase functionalites such as including multiple eruption vents simultaneously and produce multiple masks. The former was important during the vent migration phase, while the latter was necessary to communicate uncertainty in model results.

Future improvements of the code and developed approaches include strategies to make (i) quantitative evaluation strategy of modelling results that can be used during the crisis (ii) establish uncertainty testing of input parameters and (iii) make map templates to efficiently disseminate results.

The lava flow model results were shared regularly with the scientific community, the agencies responsible for the operations in-situ and to the general public (through news articles on institutions websites). Regardless the multiple issues, challenges

and open questions listed in this review, the numerical code and the established modelling procedures were considered very successful for responding to an eruption that called tourists and visitors from all over the world.

## 7 Code and Data availability

MrLavaLoba is code was published in 2018 (de' Michieli Vitturi and Tarquini, 2018) and is freely available at the model repository (http://demichie.github.io/MrLavaLoba/). Relevant data for this study can be found in Pedersen et al. (2022b),

https://doi.org/10.5281/zenodo.6581470. The outputs of MrLavaLoba simulations are available upon request to the corresponding author.

## Author contributions

GBMP undertook manuscript preparation, model implementation, strategy development, validation, and formal analysis, and communication with stakeholders. MAP undertook the background research, model implementation for pre-eruption runs, formal analysis, communication with stakeholders and review of manuscript. SB undertook the background research,



supervised strategy development and stakeholder contact and review of the manuscript. ST and MdMV developed the code, supervised strategy development and code implementation and reviewed the manuscript. BÓ undertook the background

research and reviewed the manuscript and RHÞ contributed to map production and review of the manuscript.

## Competing interests

The contact author has declared that neither they nor their co-authors have any competing interests.

## Acknowledgements

The authors would like to acknowledge scientist and stakeholders providing feedback on the communication of hazard maps
during stakeholder meetings. Furthermore, they would like to thank their financial support.

## Financial support

This research has been supported by the Icelandic Research fund, Grant No. 206755-052 awarded to GBMP.

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






**Figure 1: a) Overview of the Fagradalsfjall area at the end of the eruption. Vents are marked with dots and numbered chronologically after opening time. Lava thickness map is from September 30, 2021 (Pedersen et al., 2022b). Dashed black box indicate the extent of frame b.  b)  Overview of the Fagradalsfjall area by the end of the Phase 1. Lava thickness map is from April 5, 2021 (Pedersen et al., 2022b). Hiking paths are in dashed blue. c) Map of the Reykjanes Peninsula. The red box indicates the area displayed in a). Densely populated areas are marked in gray. Volcanic systems (Sæmundsson and Sigurgeirsson, 2013) are marked with orange and denoted by capital letter according to their name; R: Reykjanes, S: Svartsengi, F: Fagradalsfjall, K: Krýsuvík, B: Brennisteinsfjöll, H: Hengill. Background topography is based on the IslandsDEM (Porter et al., 2018).**

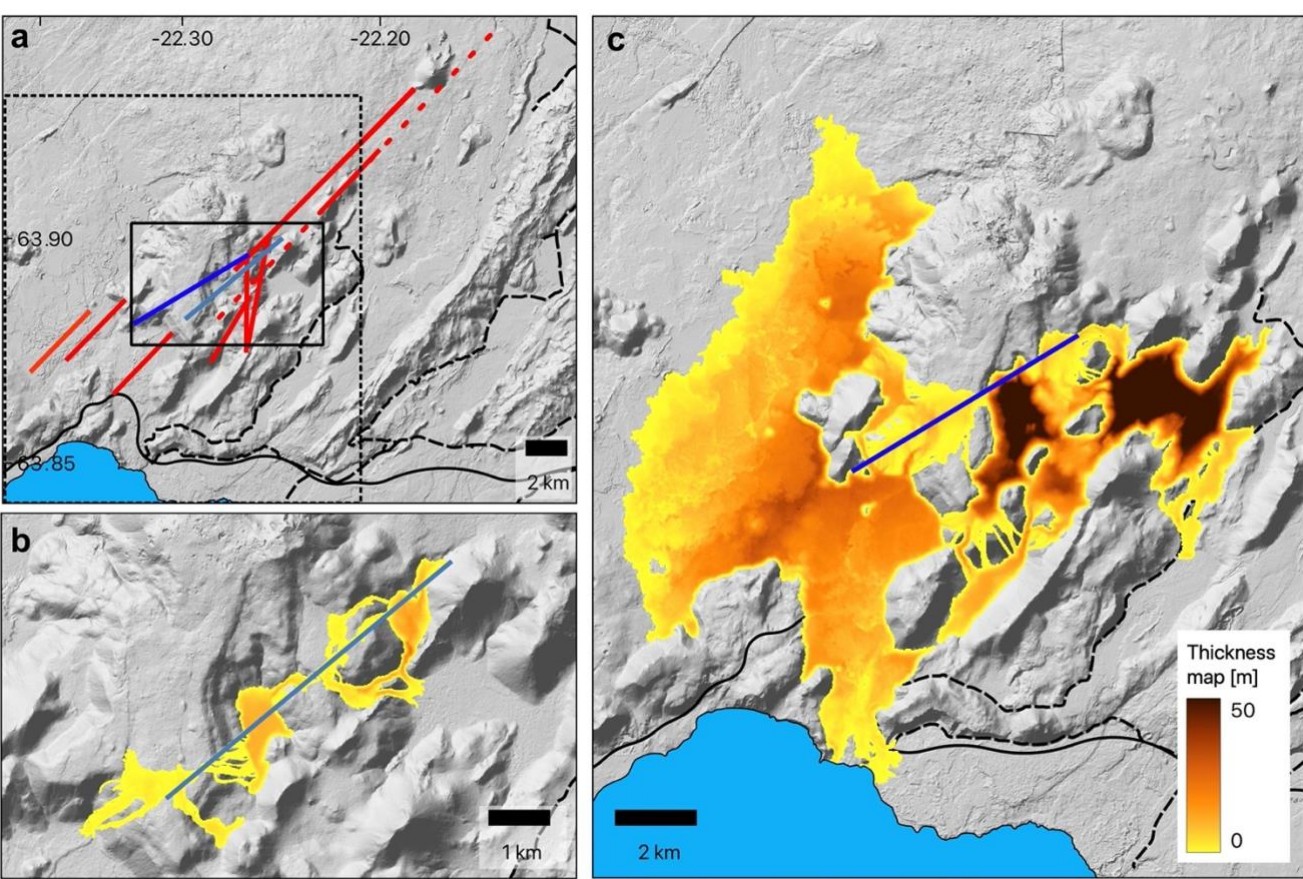


**Figure 2: (a) Overview of pre-eruption modeled potential fissures. Solid black box indicates the extent of frame b and dashed black box indicates the extent of frame c. (b) Worst-case scenario (300 m³/s) for t= 180 min for a pre-eruption fissure (dark blue).  (c) Moderate scenario (0.3 km³) run for a pre-eruption fissure (light blue). Background topography is based on the IslandsDEM (Porter et al., 2018).**

**Figure 3: Comparison between thickness maps obtained from the lava field (Upper box) in the first 24 days of the eruption (Pedersen et al., 2022b) and lava simulation thickness maps (Lower box). The volume of each thickness map is noted above each map. Thickness scale is the same for DDEMs and lava simulation thickness maps. The initial length of the first fissure is marked as**
**a black line. The last two frames showing thickness maps from the eruption extend into phase 2 and therefore also include the fissures that opened up in in phase 2. Background topography is based on the IslandsDEM (Porter et al., 2018).**



**Figure 4: Comparison between thickness maps obtained from the lava field (Pedersen et al., 2022b) from April 5 to May 3 (Upper box) and lava simulation thickness maps performed during the vent migration phase (Lower box). The lava simulations in the bottom panel show the cumulative thickness of the input thickness map and the results from the simulation. The extent of the thickness maps used as input to the simulation has a solid black outline, while the extent of the lava thickness maps from the lava field with comparable volume is marked with hatched outlines. The active vents are in white, except for the fissure 2, where the initial two fissure segments are shown as black lines. The full extent of the initial fissure segments was used to for the lava flow simulations. Background topography is based on the IslandsDEM (Porter et al., 2018).**


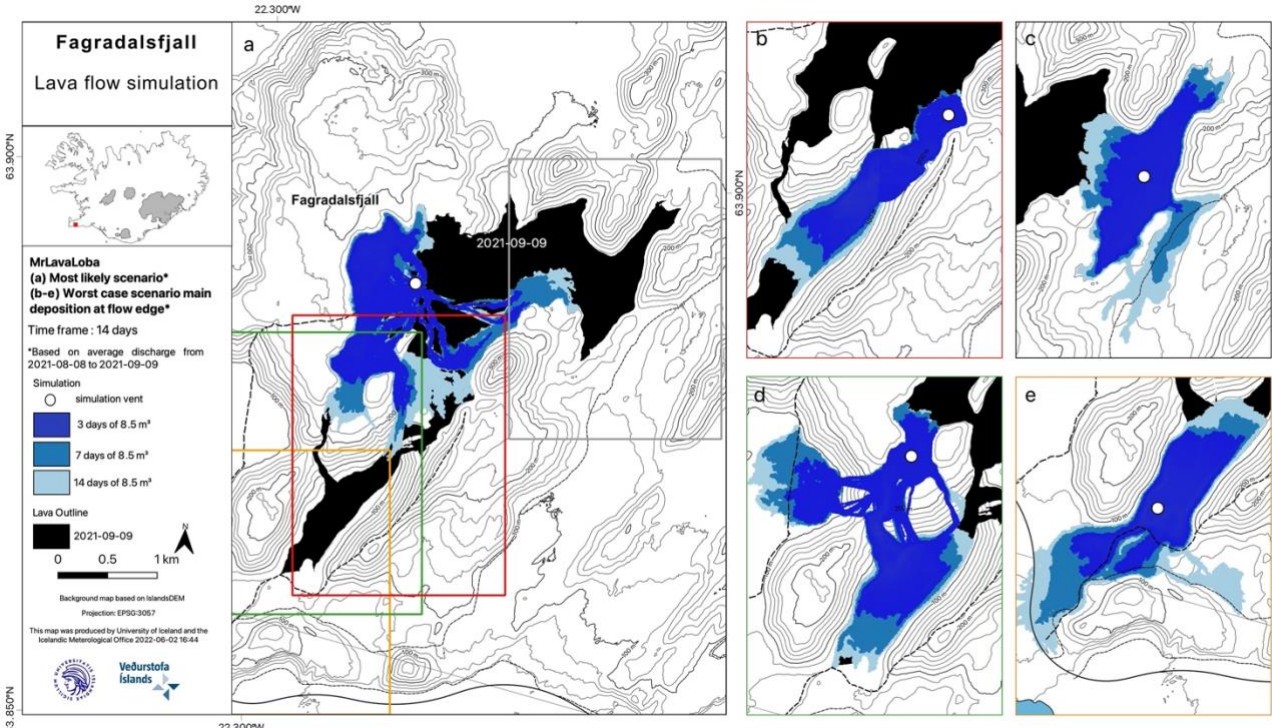

**Figure 5: An example of how the short-term worst-case scenarios from 9 September, 2021 were presented. (a) Simulation from vent 5, which was considered the most likely scenario (b) worst-case scenario for Syðri-Meradalur to investigate if lava could exit the valley through the saddle point to the southeast. Based on these results the lava seemed more likely to exit south to Nátthagi (c)**
**worst-case scenario for Meradalir to investigate if lava would spill to the east out of the valleys. Based on the results this was considered an option given the vast majority of erupted volume would reach Meradalir. (d) worst-case scenario for Geldingadalir to investigate if lava would spill to the southwest into Nátthagikríka over the build barriers. Based on the results this was considered an option. (c) worst-case scenario for Nátthagi to investigate if lava would spill to the south out of the valley. Based on the results this was considered an option given the vast majority of erupted volume would reach Nátthagi. Background**
**topography is based on the IslandsDEM (Porter et al., 2018).**


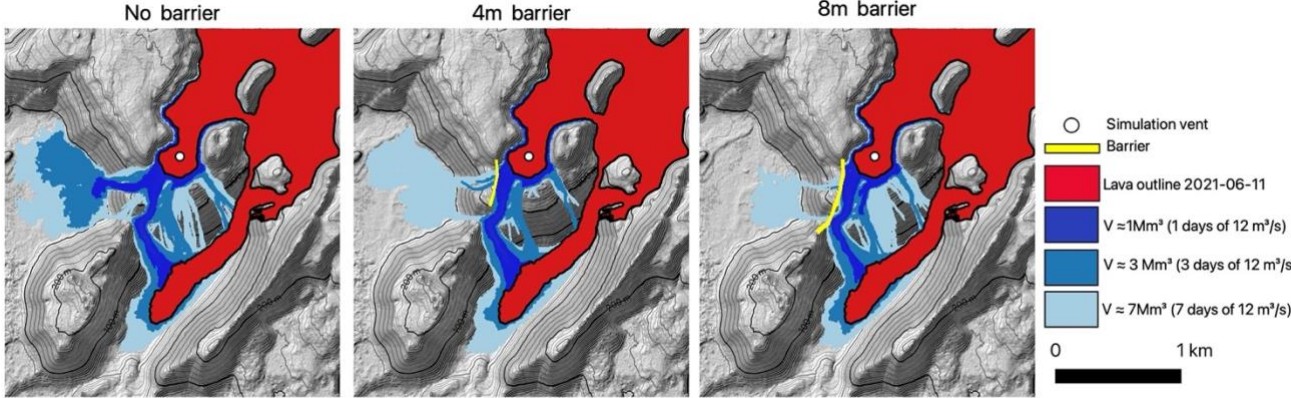

**Figure 6: An example of lava simulations predict that Geldingadalir would overflow with and without lava barriers. The simulation vent is located in southern Geldingadalir based on observation of lava inflation in this area. Data based on survey from June 11, 2021 and the calculated volumes and time scales are based on the estimated TADR from that survey. Background topography is based on the IslandsDEM (Porter et al., 2018).**

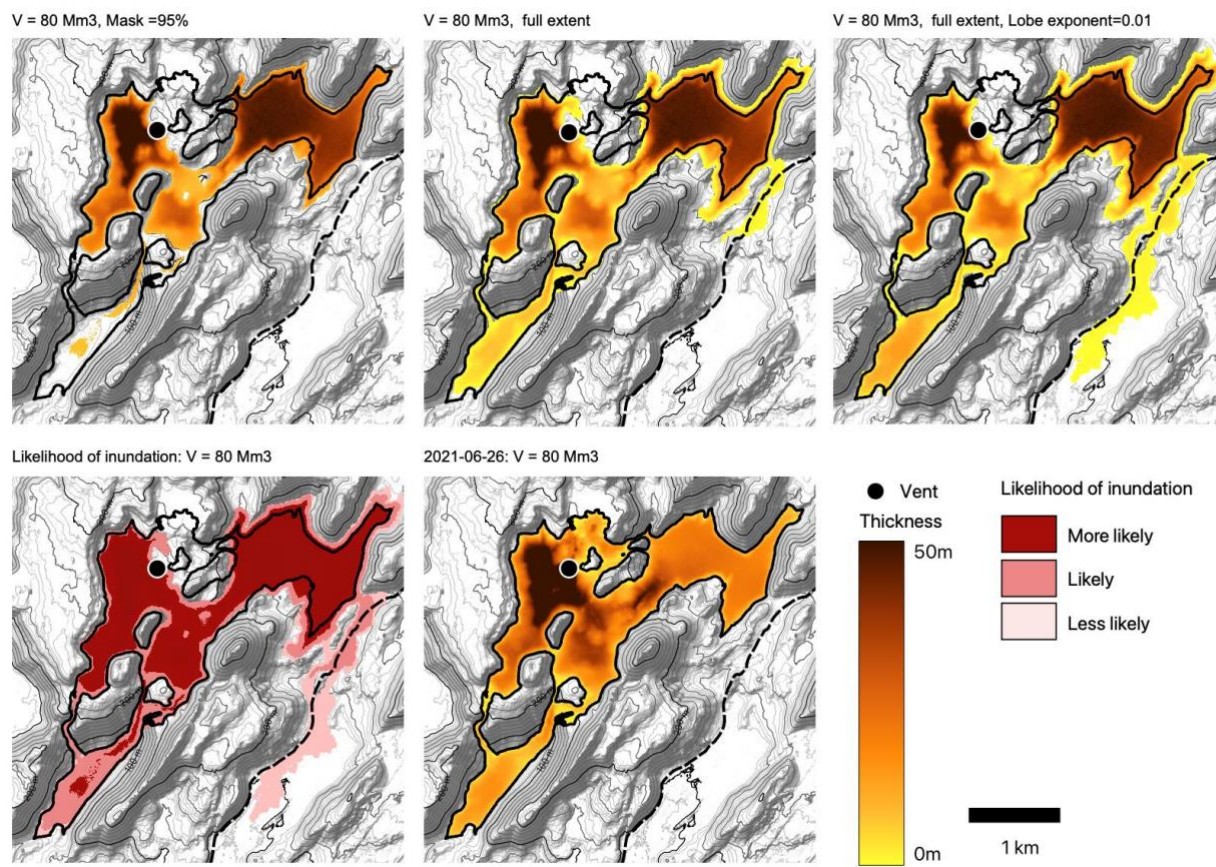

**Figure 7: Top row: Example of the best tuning result for the 80 Mm3 simulation showing the 95 % lava thickness mask representing the category "More likely" and the full extent representing "Likely" category. To the right is a 80 Mm3 simulation result with a lower lobe exponent, which represent the "Less likely" category. Bottom row: Map showing the likelihood of inundation based on the 80 Mm3 simulations shown in the top row (Left) and lava thickness map from 2021-06-26 (Pedersen et al., 2022b), which can be compared to the simulation results in the top row (Right). Background topography is based on the**
**IslandsDEM (Porter et al., 2018).**


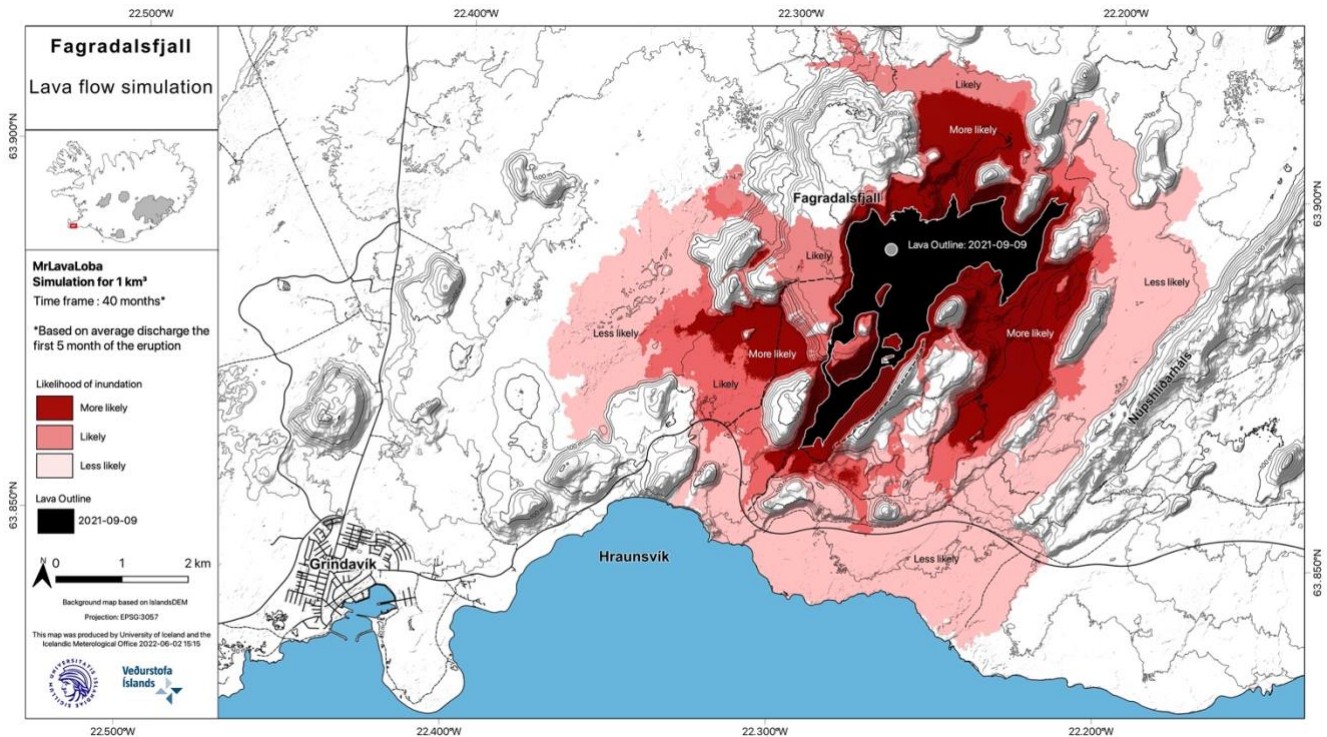

**Figure 8: Long-term scenario for lava emplacement of a volume of 1000 Mm³ issued from vent 5. Background topography is based on the IslandsDEM (Porter et al., 2018).**




**Table 1 Overview of implementation strategies of MrLavaLoba for pre- and syn-eruptive simulations**


| | Pre-eruptive simulations | | | Syn-eruptive simulations | | | |
|---|---|---|---|---|---|---|---|
| | Short -term | Longer-term | | Short -term | | | Longer-term |
| | Worst-case scenario | Small eruption scenario | Moderate eruption scenario | Phase 1: Geldingadalir | Phase 2:Vent migration | Phase 3-5: Compound lava field | Phase 3-5: Compound lava field |
| **Goal** | Assess likely areas inundated hours after eruption start of high-effusion rate eruption | Gain insight to areas likely to be inundated within weeks from eruption start | Gain insight to areas likely to be inundated months after eruption start | Assess likely areas inundated first weeks of the eruption, including when Geldingadalir would spill into Syðri-Meradalur | Assess likely areas inundated during vent migration period, including when Geldingadalir would spill into Syðri-Meradalur | Forecast spilling of one valley to another. Evaluate areas endangered to lava inundation, specifically areas close to hiking paths/safety zones. Evaluate when barriers may be compromised | Address infrastructure endangered to lava flow inundation on month to year scale |
| **Approach** | Multiple runs with with various volumes simulating V= Q*t, where Q= 300m3/s. Some of the input variables are made time-dependent | One run based on a pre-defined parameter space derived from scenarios compared with historical lava flows | One run based on a pre-defined parameter space derived from scenarios compared with historical lava flows | Multiple runs with with various volumes simulating V= Q*t, where Q= TADR measurements from photogrammetric surveys. Some of the input variables are made time-dependent | Multiple runs with with various volumes simulating V= Q*t, where Q= TADR measurements from photogrammetric surveys. Some of the input variables are made time-dependent | Multiple runs with with various volumes simulating V= Q*t, where Q= TADR measurements from photogrammetric surveys. Some of the input variables are made time-dependent | Multiple runs with with various volumes () simulating V= Q*t, where Q= TADR_mean for the eruption . All input parameters were tuned based on lava thickness maps obtained in June having volumes between 53-80 Mm3 |
| **Evaluation** | Lava run out length + lava thickness | Tuning compared to Illahraun (historical scenario) | Tuning compared to Arnaseturhraun (historical scenario) | Qualitative comparison to lava thickness maps | Qualitative comparison to lava thickness maps | NA | The eruption stopped at V= 150 Mm3 and thus not comparable to our smallest long-term scenarios (250Mm3) |
| **DEM** | Pre-eruption DEM (5 m x 5 m ) | Pre-eruption DEM (10 m x 10 m ) | Pre-eruption DEM (10 m x 10 m ) | Pre-eruption DEM (5 m x 5 m) | Pre-eruption DEM (2 m x 2 m)/ Pre-eruption DEM (5 m x 5 m) + DDEM from 2021-04-12 (5 m x 5 m) / Pre-eruption DEM (5 m x 5 m) + DDEMfrom 2021-04-21 (5 m x 5 m) | Pre-eruption DEM + newest thickness map + thickness map of lava barriers (if relevant) | Pre-eruption DEM (10 m x 10 m ) + newest thickness map + thickness map of lava barriers (if relevant) |





| Vents | 2/ 10 km fissures selected in the area of deformation | 2/ 10 km fissures selected in the area of deformation | 2/ 10 km fissures selected in the area of deformation | 180 m fissure. Location and extent after the maximum opening observed in Geldingadalir | F2/All vents/ 5 southernmost vents | lava front edge at locations | Vent 5 |
|---|---|---|---|---|---|---|---|
| **Variable Parameters** | | | | | | | |
| **t [min]** | 30, 60, 180 , (360, 720 ) | NA | NA | t= V/ 10m3/s | t= V/ 5m3/s / t= V/ 8m3/s / t= V/ 5.1m3/s | 3, 7, 14 days. If 3 day scenario revealed inundation of areas close to hiking paths/safety zones 6 hr, 12 hr and 24 hr scenarios would be modelled as well | t= V/ TADR_mean |
| **Volume [m3]** | 300 m3/s * t*60s | 0.02 | 0.3 | 0 - 10 Mm3 | 0 - 2.6 Mm3 /0 - 3 Mm3/ 0 - 6 Mm3/ | TADR m3/s * t*60s | 150 Mm3, 250 Mm3, 350Mm3, 500 Mm3, 1000Mm3, 5000 Mm3 |
| **n flows** | 150 per km fissure | 400 | 1600 | 10 | 10 / 80/80 | 160 | 2000 /(3000 for km3) |
| **min_n_lobes** | 3.33 * t | 400 | 1500 | 2 * t[min] | 2 * t[min] / 2 * t[min] / 1 * t[min] | 1*t [min] | 3500 |
| **Lobe exponent** | 0.07 | 0.03 | 0.015 | 0.07 | 0.03/ 0.07/ 0.07 | 0.05 | 0.02 / 0.01 |



**Table 2: Overview of advantages and disadvantages of the MrLavaLoba code**

| MrLavaLoba code | |
|---|---|
| **Advantages** | **Disadvantages** |
| <ul><li>Free & easy to run in python</li><li>Very flexible and can be used for various simulation purposes</li><li>Fast computational time</li><li>Can run very large scenarios</li><li>Change topography during model run</li><li>Produce lava thickness layer</li><li>Includes volume + final extent</li><li>Can be used to assess infilling of depressions, valleys and overflows of barriers</li><li>Can handle multiple vents</li><li>Easy to implement barriers & new topography/thickness layer</li></ul> | <ul><li>Not a physical model: input parameters have to be tuned to: known scenarios; individual eruptions; specifice to type of activity; and different topography</li><li>Do not provide time/velocity evolution of lava emplacement</li><li>Do only mimic channel/tube formation if tuned for that,- does not develop inherently in the model</li><li>Do not include vent processes→ underestimate thickness of deposits close to the vent (thus DDEM should be implemented to account for that)</li><li>Results not designed for hazard communication</li></ul> |






## Appendix

Table A1: Overview of all input parameters used depending on modelling approach

| Tuning Parameters | Pre-eruptive simulations | | | Syn-eruptive simulations | | | |
|---|---|---|---|---|---|---|---|
| | Short -term | Longer-term | | Short -term | | | Longer-term |
| | Worst-case scenario: 300 m3/s | Small eruption scenario | Moderate eruption scenario | Phase 1: Geldingardalir | Phase 2:Vent migration | Phase 3-5: Compound lava field | Phase 3-5: Compound lava field |
| source DEM [in EPSG 3057] | Pre-eruption DEM (5 m x 5 m ) | Pre-eruption DEM (10 m x 10 ) | Pre-eruption DEM (10 m x 10 m ) | Pre-eruption DEM (5 m x 5 m) | Pre-eruption DEM (2 m x 2 m)/ Pre-eruption DEM (5 m x 5 m) + DDEM from 2021-04-12 (5 m x 5 m) / Pre-eruption DEM (5 m x 5 m) + DDEMfrom 2021-04-21 (5 m x 5 m) | Pre-eruption DEM + newest thickness map + thickness map of lava barriers (if relevant) | Pre-eruption DEM (10 m x 10 m ) + newest thickness map + thickness map of lava barriers (if relevant) |
| vent_flag | 2 | 2 | 2 | 2 | 8 | 8 | 8 |
| x_vent [in EPSG 3057] | [x1, x2] | [x1, x2] | [x1, x2] | [339326 ,339423 ] | Combination of following vents: [339366, 339386, 339522, 339489, 339467, 339473, 339545] | [339048] | [339048] |
| y_vent [in EPSG 3057] | [y1, y2] | [y1, y2] | [y1, y2] | [380202 ,380364 ] | Combination of following vents: [380288, 380319, 380637, 380515, 380471, 380440, 380695] | [380058] | [380058] |
| hazard_flag | 1 | 1 | 1 | 1 | 1 | 1 | 1 |
| fissure_probabilities | NA | NA | NA | NA | [1, 1, 1, 5, 5, 5, 1] | [1] | [1] |
| masking_threshold | 0.96 | 0.96 | 0.96 | 0.96 | 0.96 | [0.68 ,0.96] | 0.95 |
| n_flows | 150 per km fissure | 400 | 1600 | 10 | 10 / 80/80 | 160 | 1000 |
| min_n_lobes | 3.33 * t[min] | 400 | 1500 | 2 * t[min] | 2 * t[min] / 2 * t[min] / 1 * t[min] | 1*t [min] | 3500 |
| max_n_lobes | min_n_lobes | min_n_lobes | min_n_lobes | min_n_lobes | min_n_lobes | min_n_lobes | min_n_lobes |
| volume_flag | 1 | 1 | 1 | 1 | 1 | 1 | 1 |
| total_volume | 300 m3/s*time | 20000000 | 300000000 | TADR [m3/s] * t[s] | TADR [m3/s] * t[s] | TADR [m3/s] * t[s] | 150 M, 250 M, 350M, 500 M, 1000M, 5000M |
| fixed_dimension_flag | 1 | 1 | 1 | 1 | 1 | 1 | 1 |
| lobe_area | 250 | 1000 | 1000 | 250 | 250 | 250 | 1000 |
| thickness_ratio | 0.9 | 2 | 2 | 0.9 | 0.9 | 0.9 | 2 |
| topo_mod_flag | 2 | 2 | 2 | 2 | 2 | 2 | 2 |
| n_flows_counter | 1 | 1 | 1 | 1 | 1 | 1 | 1 |




| n_lobes_counter | 1 | 1 | 1 | 1 | 1 | 1 | 1 |
|---|---|---|---|---|---|---|---|
| thickening_parameter | 0.06 | 0.06 | 0.06 | 0.06 | 0.06 | 0.06 | 0.02 |
| lobe_exponent | 0.07 | 0.03 | 0.015 | 0.07 | 0.07 | 0.05 | 0.02 |
| max_slope_prob | 0.8 | 0.8 | 0.8 | 0.8 | 0.8 | 0.8 | 0.8 |
| inertial_exponent | 0.1 | 0.1 | 0.1 | 0.1 | 0.1 | 0.1 | 0.1 |
