# Peer review of "Lava flow hazard modelling during the 2021 Fagradalsfjall eruption, Iceland: Applications of MrLavaLoba"

_Natural Hazards and Earth System Sciences, 2022_

## Referee Comment (RC1)

**Lava flow hazard modelling during the 2021 Fagradalsfjall eruption, Iceland: Applications of MrLavaLoba**

Gro B. M. Pedersen1, Melissa A. Pfeffer2, Sara Barsotti2, Simone Tarquini3, Mattia de´ Michieli Vitturi3,4, Bergrún Óladóttir2, Ragnar Heiðar Þrasstarson2

5

[referee-annotated manuscript omitted]

---

## Author Comment (AC1)

[revised manuscript text omitted]

• Fast computational time
• Can run very large scenarios
• Change topography during model run
• Produce lava thickness layer
• Includes volume + final extent
• Can be used to assess infilling of depressions, valleys and overflows of barriers
• Can handle multiple vents
• Easy to implement barriers & new topography/thickness layer | • Not a physical model: input parameters have to be tuned to: known scenarios; individual eruptions; specifice to type of activity; and different topography
• Do not provide time/velocity evolution of lava emplacement
• Do only mimic channel/tube formation if tuned for that,- does not develop inherently in the model
• Do not include vent processes→ underestimate thickness of deposits close to the vent (thus DDEM should be implemented to account for that)
• Results not designed for hazard communication |

[Figure]

[Figure]

**Appendix**

Table A1: Overview of all input parameters used depending on modelling approach

| Tuning Parameters | Pre-eruptive simulations | | | Syn-eruptive simulations | | | |
|---|---|---|---|---|---|---|---|
| | Short -term | Longer-term | | Short -term | | | Longer-term |
| | Worst-case scenario: 300 m3/s | Small eruption scenario | Moderate eruption scenario | Phase 1: Geldingardalir | Phase 2:Vent migration | Phase 3-5: Compound lava field | Phase 3-5: Compound lava field |
| source DEM [in EPSG 3057] | Pre-eruption DEM (5 m x 5 m ) | Pre-eruption DEM (10 m x 10 ) | Pre-eruption DEM (10 m x 10 m ) | Pre-eruption DEM (5 m x 5 m) | Pre-eruption DEM (2 m x 2 m)/ Pre-eruption DEM (5 m x 5 m) + DDEM from 2021-04-12 (5 m x 5 m) / Pre-eruption DEM (5 m x 5 m) + DDEMfrom 2021-04-21 (5 m x 5 m) | Pre-eruption DEM + newest thickness map + thickness map of lava barriers (if relevant) | Pre-eruption DEM (10 m x 10 m ) + newest thickness map + thickness map of lava barriers (if relevant) |
| vent_flag | 2 | 2 | 2 | 2 | 8 | 8 | 8 |
| x_vent [in EPSG 3057] | [x1, x2] | [x1, x2] | [x1, x2] | [339326 ,339423 ] | Combination of following vents: [339366, 339386, 339522, 339489, 339467, 339473, 339545] | [339048] | [339048] |
| y_vent [in EPSG 3057] | [y1, y2] | [y1, y2] | [y1, y2] | [380202 ,380364 ] | Combination of following vents: [380288, 380319, 380637, 380515, 380471, 380440, 380695] | [380058] | [380058] |
| hazard_flag | 1 | 1 | 1 | 1 | 1 | 1 | 1 |
| fissure_probabilities | NA | NA | NA | NA | [1, 1, 1, 5, 5, 5, 1] | [1] | [1] |
| masking_threshold | 0.96 | 0.96 | 0.96 | 0.96 | 0.96 | [0.68 ,0.96] | 0.95 |
| n_flows | 150 per km fissure | 400 | 1600 | 10 | 10 / 80/80 | 160 | 1000 |
| min_n_lobes | 3.33 * t[min] | 400 | 1500 | 2 * t[min] | 2 * t[min] / 2 * t[min] / 1 * t[min] | 1*t [min] | 3500 |
| max_n_lobes | min_n_lobes | min_n_lobes | min_n_lobes | min_n_lobes | min_n_lobes | min_n_lobes | min_n_lobes |
| volume_flag | 1 | 1 | 1 | 1 | 1 | 1 | 1 |
| total_volume | 300 m3/s*time | 20000000 | 300000000 | TADR [m3/s] * t[s] | TADR [m3/s] * t[s] | TADR [m3/s] * t[s] | 150 M, 250 M, 350M, 500 M, 1000M, 5000M |
| fixed_dimension_flag | 1 | 1 | 1 | 1 | 1 | 1 | 1 |
| lobe_area | 250 | 1000 | 1000 | 250 | 250 | 250 | 1000 |
| thickness_ratio | 0.9 | 2 | 2 | 0.9 | 0.9 | 0.9 | 2 |
| topo_mod_flag | 2 | 2 | 2 | 2 | 2 | 2 | 2 |
| n_flows_counter | 1 | 1 | 1 | 1 | 1 | 1 | 1 |

[Figure]

[Figure]

| | | | | | | | |
|---|---|---|---|---|---|---|---|
| n_lobes_counter | 1 | 1 | 1 | 1 | 1 | 1 | 1 |
| thickening_parameter | 0.06 | 0.06 | 0.06 | 0.06 | 0.06 | 0.06 | 0.02 |
| lobe_exponent | 0.07 | 0.03 | 0.015 | 0.07 | 0.07 | 0.05 | 0.02 |
| max_slope_prob | 0.8 | 0.8 | 0.8 | 0.8 | 0.8 | 0.8 | 0.8 |
| inertial_exponent | 0.1 | 0.1 | 0.1 | 0.1 | 0.1 | 0.1 | 0.1 |

---

## Author Response (AR1)

Dr. Giovanni Macedonio,
Editor
Natural Hazards and Earth System Sciences

Dear Giovanni Macedonio,

Please find enclosed the revised manuscript submitted for Natural Hazards and Earth System Sciences entitled "Lava flow hazard modelling during the 2021 Fagradalsfjall eruption, Iceland: Applications of MrLavaLoba" by Gro B. M. Pedersen, Melissa A. Pfeffer, Sara Barsotti, Simone Tarquini, Mattia de´ Michieli Vitturi[3], Bergrún A. Óladóttir and Ragnar Heiðar Þrastarson.

The manuscript has been thoroughly revised and benefitted significantly from reviewer 1 and the editor and the authors would like to thank them for thorough, and helpful comments.
Overall, the changes can be divided into changes of the manuscript and changes of figures and tables. These changes are based on the general and detailed comments from reviewer 1 (see in detail below):

Changes to the manuscript:
As a response to reviewer 1 the manuscript has been through many iterations to shorten and simplify the main text, which is 100 lines shorter (approximately 3 pages) and is now 540 lines. As can be seen in the manuscript with track changes all sections have been thoroughly revised to ensure that the scientific motivations for the different modelling approaches during this changeable 6-month long eruption are conveyed better.

Furthermore, a new appendix B has been added providing technical information on the code changes implemented during and post 2021 Fagradalsfjall eruption.

Figure & table changes:
In response to reviewer 1 new figures were added to help the reader and to condense the text. This includes:
New Figure 2: Illustration of the temporal development of the volcanic unrest, the simulations performed addressing these developments and the identified simulation challenges.
New Figure 3: Flow chart for the lava flow simulations performed during the Fagradalsfjall 2021 volcanic unrest.
Table 1, 2, A1 was revised for spelling mistakes and consistency.

General comments:
Response to general comments from reviewer 1 (RS 1) can be found below. The authors have responded to the general comments from RS1 section by section. The bold text is comments made by RS1 and the regular text is our response to the comment above.

Detailed response
Since the NHESS system does not allow to have more than file uploaded under authors response, the authors will refer to the detailed response to reviewer 1 has previously been submitted to the journal.

As the corresponding author for this manuscript, I can be reached at the following address:

Gro B. M. Pedersen
School of Engineering and Natural Sciences
University of Iceland
Sturlugata 7, Askja
102 Reykjavík
Iceland
E-mail: gro@hi.is
Phone: + 354 8232368

Thank you for your consideration of this contribution.

Best regards,
Gro Birkefeldt Møller Pedersen

RC1: 'Comment on nhess-2022-166', Anonymous Referee #1, 07 Dec 2022:
Citation: https://doi.org/10.5194/nhess-2022-166-RC1

**The paper is a chronological narration of how the software MrLavaLobe was used and modified during the 2021 Fagradalsfjall eruption occurred in the Reykjanes peninsula, Iceland. The manuscript is really interesting, and some sections are also compelling.**

Thanks.

**But the text has some problems. In the manuscript, the authors initially describe the eruption, referring to a work in press, and this description seems a summary of the cited work.**

It is correct that we in the initial manuscript only reference the Pedersen et al. (in press) paper for the eruption evolution. Upon submission this was the only paper "in press" describing this eruption and is therefore the only paper cited. This has now been improved because more papers have been published (see section 2 and 2.1).

**Then they describe the software without focusing on input parameters and procedures but describing the functionality with a qualitative approach.**

We choose to cite the paper (de' Michieli Vitturi and Tarquini, 2018) which was published with the release of the software instead of describing the technical details in this paper. We want to emphasize the usage of this code during the Fagradalsfjall 2021 eruption. In table 1 we do provide an overview of procedures applied, their goal, the approach and key input parameters. The purpose of having these together in a table is exactly to describe the functionality in a qualitative way that enables the reader to both understand the motivation and process and to be able to duplicate our work. Table A1 (in the appendices) provides all input parameters (or input parameter ranges). We do agree that the code changes made during this eruption must be described better, and not only in the qualitative fashion as done in this manuscript. We have therefore added appendix B describing the exact changes in the code. We have also done a better job of referring to the tables, so this is much clearer for the reader.

**Then narrate the eruptive crisis and the results of the software used in real time. Finally, they describe the modification and the improvements to the software that were necessary to communicate with the stakeholders and predict lava inundation in a future eruption. The manuscript is too long and, while reading, the scientific motivations of the work are lost.**

We do agree that the initial manuscript was too long and it has been shortened, repetitions have been deleted, and the text is now more concise ensuring that the motivation for the different modelling approaches is conveyed better. This has partly been done by adding a new figure (New fig. 2) that show the evolution of the eruption, how the different activity phases and styles required different approaches to the modelling and what we did to achieve our objectives (code changes or changes in how the code was implemented and/or results were displayed). We have also made sure that figures and tables are better referenced since lots of information that RS1 requests can be found there (especially in the tables).

**The manuscript also lacks a clear introduction to the software and this lack impedes a full understanding of the reported improvements if not by reading the original paper and making comparison with this one.**

We have improved the introduction to the software in section 3 and provided more technical information about software changes in appendix B. However, the goal in this manuscript is to apply an already published code to the Fagradalsfjall 2021 eruption and we therefore think it is logical that the reader may need to read the de' Michieli Vitturi and Tarquini (2018) to get the full overview of what the code does. Otherwise, this already too long paper gets even longer, and we think that our goal of the paper to show the usage of the code gets further lost in describing the code itself. However, we have made sure that all changes made in the code during this volcanic crisis (and thus changes in the code from when it was published in 2018) will be described better, both in the paper itself and with substantial detail in appendix B.

**I found the manuscript having a journalistic approach, not suitable for a scientific journal. Otherwise, if the authors intended to publish a technical report about the software improvements due to a real time application, then they should remove the chronological description of the happenings and concentrate on the technical improvements of the software and its application to the 2021 Fagradalsfjall eruption.**

We think this critique derives from our goals for the paper not coming through well enough. This paper is not intended to be a technical report focusing on the code changes, - but rather it should be addressing how evolving lava hazards of the Fagradalsfjall 2021 eruption led to changing demands to the team providing the model results, which were addressed through changes in the MrLavaLoba code and how it was implemented. We want to highlight how MrLavaLoba code can be used in an ongoing effusive eruption, to show all the novel aspects of this interplay between the needs of end users and the potential of the scientific tool that has not been described before. Furthermore, we wish to highlight the caveats as of now and improvements that should be emphasized in the future. We will make sure that these goals are clearer in the abstract, introduction and conclusion of this paper, so the reader is clear on the goals of the paper.

The authors think the chronical narrative is necessary since the changing in activity led to different ways of implementing the code and to code changes. The way this was done, was very much a consequence of the eruption evolution and doing this during this eruption. Many of the tasks we were trying to address to provide information to the hazard assessment would have been done differently post-eruption and many of them we could not have imagined in the pre-eruption phase prior to the Fagradalsfjall 2021 eruption. Since the eruption lasted half a year and the activity changed multiple times during this eruption, it is obviously a complex task and we tried to convey this message better in the new version of the manuscript.

**I appreciated the paper that is well organized in sections but it is too long and the text gets lost in long explanations that could be summarized and made simpler. Moreover, in the manuscript I found some repeated sentences that authors should eliminate. The English is somewhere not fluent and I requested to rewrote some sentences. Sentences are often too long and dispersive; verbs are somewhere used in the wrong way.**

We fully agree with this statement. It is clear that our message was a bit lost in this lengthy initial manuscript and we need to improve the text. We have described numerous ways of how to do this in the detailed comments to RS1, that we also want to thank for taking the time to point out these weaknesses. We refer to the manuscript with track changes to see exactly how this has been done, since substantial changes can be found in every section of the manuscript.

**I also found that figures are not correctly cited and in section 2 a figure or a citation to a figure is missing. Fig.1a and 1b are never cited in the text, while the paragraphs 2 and 2.1 need reference to figures to understand the geography and the eruptive history. The same occurs for other figures. I suggest inserting in the text the right citations of all the figures by indicating also the figure boxes useful in the text.**

We fully agree, we have improved how all figures and tables are referenced.

**I attached the pdf of the manuscript where I put my comments with suggestions and critical points, but the list is not exhaustive. I think that the authors should do an effort to re-reading the manuscript and re-writing the longer and twisted sentences and eliminate repetitions. The authors should also rethink the qualitative setting given to their paper substituting the long descriptive part with short quantitative sentences. For these reasons I suggest a major revision.**

Thanks again for these comments, we have gone through them all agree with many of them. Those where we disagree, we have explained why. We have shortened the text substantially, deleted repetitions, and improved the language.

---

## Referee Report (RR1)

**Review of the manuscript No. NHESS-2022-166**

"*Lava flow hazard modelling during the 2021 Fagradalsfjall eruption, Iceland: Applications of MrLavaLoba*" by Pedersen et al.

This manuscript deals with the numerical modelling of lava flow before and during the 2021 Fagradalsfjall volcanic eruption in Iceland. I see that the manuscript was significantly changed based on the reviewer's comments: it is shortened, many issues have been clarified, references have been added. Meanwhile, I have a few structural issues and a few scientific points to be clarified before the manuscript could be accepted for publication.

Structure of the manuscript.

- Abstract, Introduction and Conclusion contain identical sentences, which should be avoided.
- The abstract must also contain a sentence or two sentences that describes the main findings of the modelling. It should also contain elements describing the context of your work as well as a sentence comparing your work to existing knowledge explaining the importance of your work in the context of disaster management.
- The Introduction section should contain more than it is now presented (2 paras). It should state about the topic of the manuscript and why it is important to the natural hazards' community, to the advancement of knowledge, and to society. It should introduce the region under the study and present a summary of the state-of-art knowledge about lava flow in the region. The authors should cite the publications that initiated the main debate or posed the main questions, which are intended to be addressed. The authors should not limit themself to their own papers on the subject or to those of their close collaborators, and to give proper credit to all opinions/ideas published on the subject (e.g., operational lava flow modelling). Such description of the works done earlier will allow the authors to position your finding in the Discussion section (based on the modelling) with respect to what is known/done before this study. The authors should also state the outstanding questions that need to be addressed on the topic of your paper, the essential data or constraints that need to be collected. Partly this is done, but structurally, the paper jumps from one topic to another. Therefore, I would propose to consider the following non-significant rearrangement of the sections.
- Sections order:
1. Introduction (add more information as it is described above)
    1.1. Geological setting and … (current sect. 2)
    1.2. Fagra… and eruption (current sect. 2.1)
    1.3. Lava flow … (current sect. 1.1)
    1.4. Lava flow hazard … (current sect. 1.2.)
2. Data (current sect. 3.1)
    2.1. Pre-erup… (current sect. 3.1.1)
    2.2. Syn-erup… (current sect. 3.1.2)

3. Methods (current section 3.2 but drop the titles "3.2.1. Software" and "3.2.2. Implementation")

4. Results to the end of the manuscript (as they are)

Such a small restructuring will allow reading fluently without linking the pieces to each other.

Scientific (and technical) points.

- Line 53. "In a chaotic way" – better "in an unpredictable way"
- Line 54. Codes cannot be stochastic or deterministic, but the methods can. So, before the use the jargon "stochastic codes", the authors should explain what it means.
- Line 82. The authors should discuss that the discharge rate and lava temperature/radiative heat transfer should play an essential role in the lava flow pattern and flow morphology.
- Line 85. How the second part of the sentence starting from "the earlier versions" is linked to the first part of the sentence.
- Lines 170-180. How does the roughness of the computational grid (compared to the fine 2m-size DEM model) influence the results of the modelling? Different smoothing techniques applied to a fine topographic surface (to create a rough computational grid) may result in changes of the lava flow pattern.
- Lines 184-187. How essential is to write the unnecessary detail in the part of the sentence starting from "mainly following …""? Do readers need the detail, which are not explained at all, just stated? Will the details be used in this study? If not, I would delete this part of the sentence.
- Line 194. " … the lava field had become very complex to simulate." This needs to be clarified. What is the complexity of the lava field compared to pre-eruptive field? A new topographic surface? If so, is it complex? The lava fills surface valleys making the new surface topography smoother and less complex for computational aims.
- Lines 204-205. "… the number of computational flows, and)." What the number of computational flows mean? The number of the streams that the lava flow generates? How it can be pre-defined? Also, is it something missing after "and."
- Line 208. "inertial" factor should be explained.
- Line 209. What does "parent parcel" mean? Is this a branching model generating siblings parcels? Please clarify.
- Line 213. How the "final lava thickness" is determined? Clarify.
- Lines 214 (and 443). "saving masked grids obtained by considering inundated cells". Please explain the meaning of "masked grids". Does it belong to data augmentation? If so, why is it used in the modelling? Also, explain what means "inundated cells" (those cells which is covered by a modelled lava?)
- Line 216. "the probabilistic nature of the code" should be replaced "the probabilistic nature of the model".
- Line 235. "the code was optimized to accelerate runs." The optimization of the code should be briefly described and not only refer to the formal changes in the code, but also to the basic computational idea behind the modifications, e.g., faster computations based on new numerical method, rapid grid change …
- Line 262 (and elsewhere in the manuscript). Please clarify the definition of the worst-case scenarios. In which sense they are worst. From the point of hazard assessment view, some of them are not worst at all.
- Line 272. Indicate the highways in Figs. 1 and 4 (black curve? or dashed curve?). Provide explanation of the curves in the figure captures.

- Lines 294-95, 356-58, 368 (and elsewhere in the MS). "… during "emergency mode" response" (and similar sentences about meetings with the stakeholders). In Table 2, in column "Disadvantages" of the MrLavaLoba code, the authors write that "results not designed for hazard communication". If so, what do then the results of the numerical modelling mean for "emergency-mode" response? Clarify in the table and/or the text.
- Lines 374-75. "…the locations of critical lava margins were manually selected on the basis of the available knowledge and expert evaluation." Did the authors check the sensitivity of the manual selection on the inundation morphology?
- Lines 512-21. The first two paragraphs should be re-written to avoid "copy-pasted" texts from abstract and the Introduction section.
- Table 1. First column "Vents". I guess that it is the number of vents (No. vents). Correct? If so, replace it. In the current line "Vents", please clarify the sign "/" (e.g., 2/10, Fissure 2/All… 5/Critical

In conclusion, addressing the issues here is crucial for the study to meet the standards of scientific rigour and contribute to the field in a meaningful manner. Once the clarity in organization of the paper and explanation/justification are provided, the result of the study can be considered relevant for publication in NHESS.

Alik Ismail-Zadeh

Karlsruhe Institute of Technology

---

## Author Response (AR2)

Dr. Giovanni Macedonio,
Editor
Natural Hazards and Earth System Sciences

Dear Giovanni Macedonio,

Please find enclosed the revised manuscript submitted for Natural Hazards and Earth System Sciences entitled "Lava flow hazard modelling during the 2021 Fagradalsfjall eruption, Iceland: Applications of MrLavaLoba" by Gro B. M. Pedersen, Melissa A. Pfeffer, Sara Barsotti, Simone Tarquini, Mattia de´ Michieli Vitturi[3], Bergrún A. Óladóttir and Ragnar Heiðar Þrastarson.

The manuscript has been thoroughly revised and benefitted significantly from reviewer 3´s comments. First of all we want to thank the reviewer for this thorough and constructive review helping the authors to improve this manuscript. We have done our best to incorporate or answer the comments.

Main changes to the manuscript:
As a response to the comments from reviewer 3 the manuscript abstract, introduction and conclusion have partly been re-written to avoid repetition of similar sentences and the introduction has been re-arranged according to suggestions. The method section has been improved by addressing the technical questions/comments and we have changed the use of the term "stochastic code", to "stochastic model" instead. Under results we have clarified the usage of the term "worst-case scenario".

General and detailed response to comments:
Response to all general and detailed comments from reviewer 3 can be found below. The authors have responded to each comments section by section. The bold text is comments made by reviewer 3 and the regular text is our response to the comment above.

As the corresponding author for this manuscript, I can be reached at the following address:

Gro B. M. Pedersen
School of Engineering and Natural Sciences
University of Iceland
Sturlugata 7, Askja
102 Reykjavík
Iceland
E-mail: gro@hi.is
Phone: + 354 8232368

Thank you for your consideration of this contribution.

Best regards,
Gro Birkefeldt Møller Pedersen

Review of the manuscript No. NHESS-2022-166
**"*Lava flow hazard modelling during the 2021 Fagradalsfjall eruption, Iceland: Applications of MrLavaLoba*" by Pedersen et al.**

**This manuscript deals with the numerical modelling of lava flow before and during the 2021 Fagradalsfjall volcanic eruption in Iceland. I see that the manuscript was significantly changed based on the reviewer's comments: it is shortened, many issues have been clarified, references have been added. Meanwhile, I have a few structural issues and a few scientific points to be clarified before the manuscript could be accepted for publication.**

**Structure of the manuscript.**
**- Abstract, Introduction and Conclusion contain identical sentences, which should be avoided.**
We agree with this comment, and we have deleted or changed sentences that were identical or very similar.

**- The abstract must also contain a sentence or two sentences that describes the main findings of the modelling. It should also contain elements describing the context of your work as well as a sentence comparing your work to existing knowledge explaining the importance of your work in the context of disaster management.**
Thanks for this comment. We have added two sentences describing the results and how they were used. Because of the different simulation strategies implemented at different phases of the eruption, there is not one or two major findings, and because abstracts for research articles in NHESS are requested to have a length of 100–200 words, it is somewhat limited what can be described. In addition to the rephrasing to avoid repetition with introduction and conclusion we have added the following sentences:
"The model provided promising results that were shared regularly on stakeholder meetings with the monitoring personnel, scientists and civil protection representatives helping to identify potential short-term and long-term lava hazards, including evaluation of the timing of barrier overflow and the filling and spilling of lavas from one valley to another."

**- The Introduction section should contain more than it is now presented (2 paras).**
So originally the introduction was the first 2 paragraphs under section 1 as well as two sub chapters (1.1: 5 sections and 1.2: one section), and now two additional subsections have been included.

**It should state about the topic of the manuscript and why it is important to the natural hazards' community, to the advancement of knowledge, and to society.**
Thanks,- In the first section of the introduction we state "Being located in the backyard of the capital Reykjavík and the international airport, thousands of people visited the eruption each day, which therefore needed intense monitoring and thorough hazard assessment (Barsotti et al., 2023)" and ", it became a case study for the monitoring and hazard assessment for future effusive eruptions since several volcanic systems on the Reykjanes peninsula have the potential to issue lava into inhabited areas or inundate critical infrastructure." These sentences highlight why this topic was relevant to society (safety of thousands of visitors and case study for future eruptions) and the main question or task of this paper is how the lava flow hazards were addressed during the Fagradalsfjall 2021 eruption.

Furthermore, the following sentence is of interest to the natural hazard community and advancement of knowledge: "The model proved to be a useful and a flexible tool to evaluate pre-eruption as well as syn-eruptive short-term and long-term hazards during the 6-month long effusive event. Different approaches as well as new developments of the code were used to account for the changes in the eruptive behavior, and to resolve challenges provided by the complex topographic terrain, where infilling and overflowing of nested valleys created time-evolving hazards for visitors. Additionally, strategies for lava barrier testing were developed and near-real time syn-eruptive topographic models were incorporated as the eruption progressed."

**It should introduce the region under the study and present a summary of the state-of-art knowledge about lava flow in the region.**
This is in the section "Geological setting and eruptive history, which upon request from reviewer 3 now is included in section 1.1 + 1.2.

**The authors should cite the publications that initiated the main debate or posed the main questions, which are intended to be addressed. The authors should not limit themself to their own papers on the subject or to those of their close collaborators, and to give proper credit to all opinions/ideas published on the subject (e.g., operational lava flow modelling). Such description of the works done earlier will allow the authors to position your finding in the Discussion section (based on the modelling) with respect to what is known/done before this study.**
We are unsure of what the reviewer is referring to. In the section "Lava flow simulations" (now section 1.3) We start addressing different types of numerical models and then provide a short overview of operational lava flow modelling from Etna, Hawaii, Piton de la Fournaise and other lava flow modelling of other recent eruptions. Of the 35 references in this text, three are from co-authors (one of them cited twice), and another four (one cited three times) are from collaborators. We therefore don´t think we have limited ourselves to cite our own or close collaborators work. If the reviewer thinks we forgetting specific references we welcome suggestions to references in order to improve the section.

**The authors should also state the outstanding questions that need to be addressed on the topic of your paper, the essential data or constraints that need to be collected.**
As stated above the main purpose of this paper is how the lava flow hazards were addressed during the Fagradalsfjall 2021 eruption. We hope this is clear from the introduction, if not before then after the changes performed in this review including the rearrangement. The essential data is discussed under the data section.

**Partly this is done, but structurally, the paper jumps from one topic to another. Therefore, I would propose to consider the following non-significant rearrangement of the sections.**
**- Sections order:**
**1. Introduction (add more information as it is described above) 1.1. Geological se□ng and … (current sect. 2)**
**1.2. Fagra… and eruption (current sect. 2.1)**
**1.3. Lava flow … (current sect. 1.1)**
**1.4. Lava flow hazard … (current sect. 1.2.)**

**2. Data (current sect. 3.1)**

**2.1. Pre-erup… (current sect. 3.1.1)**

**2.2. Syn-erup… (current sect. 3.1.2)**
**3. Methods (current section 3.2 but drop the titles "3.2.1. So☐ware" and "3.2.2. Implementation")**
**4. Results to the end of the manuscript (as they are)**
**Such a small restructuring will allow reading fluently without linking the pieces to each other.**
Thanks, this has been changed accordingly.

**Scientific (and technical) points:**

**- Line 53. "In a chaotic way" – beter "in an unpredictable way"**
Thanks this has been changed.

**- Line 54. Codes cannot be stochastic or deterministic, but the methods can. So, before the use the jargon "stochastic codes", the authors should explain what it means.**
Thanks, we used the term "code" in a somehow improper way. In the new version, we consistently used the term "model" instead of "code" throughout the whole section 1.3 (and elsewhere, when needed), in order to remove this issue.

**- Line 82. The authors should discuss that the discharge rate and lava temperature/radiative heat transfer should play an essential role in the lava flow patern and flow morphology.**
Thanks, this is a very interesting topic and could be included in a separate chapter. However, we don´t think that such a discussion fit into the description of lava flow hazard models in Iceland and would prefer not to make this manuscript longer by adding a chapter that is not strictly necessary to understand the paper.
We have added the following description to the deterministic models, which would be the primary models that include such parameters.
That sentence now reads : "The so-called deterministic models are intended to mimic the behavior of the natural systems by modeling physical processes based on a set of conservation equations parameters including parameters such as lava discharge rate, lava temperature and heat transfer (e.g., Dietterich et al., 2017, FLOWGO: Harris and Rowland, 2001, PyFLOWGO: Chevrel et al., 2018, MAGFLOW: Cappello et al., 2016a)."

**- Line 85. How the second part of the sentence starting from "the earlier versions" is linked to the first part of the sentence.**
Thanks, we made this sentence more clear by adding "both" in the first sentence (to underline 2 models were run) and simplified the latter sentence. The sentence now reads:

"Prior to the onset of the 2014-2015 Holuhraun eruption, both VORIS ran as part of the VOLCANBOX package (https://volcanbox.wordpress.com/) within the VeTools project (http://www.evevolcanoearlywarning.eu/vetools-objectives/), and the initial versions of the MrLavaLoba model started being run (de' Michieli Vitturi and Tarquini, 2018).

**- Lines 170-180. How does the roughness of the computational grid (compared to the fine 2m-size DEM model) influence the results of the modelling? Different smoothing techniques applied to a fine topographic surface (to create a rough computational grid) may result in changes of the lava flow patern.**
The point is interesting. However, a complete exploration of the effect of DEM smoothing techniques is beyond the scope of the present work. The use of a 5 to 10 m resolution DEM for lava flow simulations is

deemed adequate [e.g. Flynn et al. 2023, now added in the reference list], and the small changes in the input DEM expected from the use of different smoothing techniques are expected to be negligible, also considering the intrinsic probabilistic nature of the MrLavaLoba model. We added the following sentence in the text "We note that the use of a 5 or 10 m cell size DEM is customary in lava flow modeling (Flynn et al. 2023), we used a 2 m cell size DEM only when a higher detail was necessary."

**- Lines 184-187. How essential is to write the unnecessary detail in the part of the sentence starting from "mainly following …""? Do readers need the detail, which are not explained at all, just stated? Will the details be used in this study? If not, I would delete this part of the sentence.**
Thanks, this sentence has been deleted.

**- Line 194. " … the lava field had become very complex to simulate." This needs to be clarified. What is the complexity of the lava field compared to pre-eruptive field? A new topographic surface? If so, is it complex? The lava fills surface valleys making the new surface topography smoother and less complex for computational aims.**
Thanks, we have clarified this by adding "due changes in the vent activity and the expansion of the lava field infilling multiple valleys at variable rates". So as described we had a vent migration phase in April, which complicated the lava flow modelling. Furthermore, the lava expansion into multiple valleys, which would be infilled at different rates at different times (basically all lava would go into one valley one week, subsequently a channel would get blocked and suddenly switching the lava into a different valley. The sentence now reads: **"(b) to update the computational domain from pre-eruption to syn-eruptive topography for the short-term simulation after April 27, when the lava field had become very complex to simulate due changes in the vent activity and the expansion of the lava field infilling multiple valleys at variable rates."**

**- Lines 204-205. "… the number of computational flows, and)." What the number of computational flows mean? The number of the streams that the lava flow generates? How it can be pre-defined? Also, is it something missing after "and."**
Thanks. The computational flows are the chains of given number of "parcels".  We have rephrased the sentence so this should be clearer:
"The MrLavaLoba code requires (a) a computational domain constituted by the pre-emplacement topography and (b) a series of input parameters (including e.g., vent(s) position and geometry, total extruded lava volume, the number of computational flows). The lava emplacement is simulated as elliptical lava"parcels" or "lobes" with a given area and thickness that are deposited step-wise along the flow path constantly modifying the topography. Each new parcel buds from an existing one, so that the simulation progresses by creating chains of a given number of "parcels".  The chains of parcels (making up a flow path) are called computational "flows".

**- Line 208. "inertial" factor should be explained.**
Thanks, we have improved the explanation of the "inertial factor", and added a reference if deeper explanation is needed. The sentence now reads:
"The direction of propagation of each parcel with respect to the parent parcel (the parcel where the budding process occurs) is determined by the local direction of the steepest descent path (computed on the topography with the addition of a random perturbation) and of an additional contribution (called "inertial factor") given by the direction of the parent parcel (de' Michieli Vitturi and Tarquini 2018)."

**- Line 209. What does "parent parcel" mean? Is this a branching model generating siblings parcels? Please clarify.**

Thanks, this has now been clarified and the sentence now reads: "The direction of propagation of each parcel with respect to the parent parcel (the parcel where the budding process occurs)…"

**- Line 213. How the "final lava thickness" is determined? Clarify.**

Thanks, this is now clarified by changing the sentence to: "The model proceeds by iteratively setting new parcels on the topography until their cumulative volume equals the lava volume prescribed for the simulation. At the end of each simulation the model provides a raster map of the final lava thickness, where each cell in the raster constitutes the cumulative thickness of all parcels that inundated that cell."

**- Lines 214 (and 443). "saving masked grids obtained by considering inundated cells". Please explain the meaning of "masked grids". Does it belong to data augmentation? If so, why is it used in the modelling? Also, explain what means "inundated cells" (those cells which is covered by a modelled lava?)**

Thanks for this comment. The masked grids are another type of result output from the model. this is now clarified by change the sentence to:

"At the end of each simulation the model provides a raster map of the final lava thickness, where each cell in the raster constitutes the cumulative thickness of all parcels that covered or inundated that cell. The model also allows saving masked grids obtained by considering inundated cells fulfilling a specified threshold value. As an example, if this threshold is set to 95% (0.95), the thinnest portion of the final lava deposit (i.e. of the map of the final lava thickness) representing 5% of the total volume is disregarded from the map, and constitute a "masked" map of the final lava deposit. This step is relevant due to the probabilistic nature of the model, where the thinnest part of the final lava deposit represents a lower probability of inundation, which may change between simulations with the same input parameters. The 95% masked grids from different simulations tend to converge (de' Michieli Vitturi and Tarquini, 2018) and represent an area "more likely" to be inundated. Through iterations of a large number of computational flows, MrLavaLoba handles the probabilistic aspect of lava emplacement (de' Michieli Vitturi and Tarquini, 2018)."

**- Line 216. "the probabilistic nature of the code" should be replaced "the probabilistic nature of the model".**

Thanks, this has changed.

**- Line 235. "the code was optimized to accelerate runs." The optimization of the code should be briefly described and not only refer to the formal changes in the code, but also to the basic computational idea behind the modifications, e.g., faster computations based on new numerical method, rapid grid change …**

Thanks, this has now been described: "Finally, the code was optimized to accelerate runs. This improvement was primarily done by providing a faster method to update the topography as new lava parcels are settled by constraining the computationally "expensive" "update topography" function to a very limited area around the new lava parcel. With respect to the version available at the beginning of the Fagradalsfjall unrest, the optimized code now is up to 5-10 times faster."

**- Line 262 (and elsewhere in the manuscript). Please clarify the definition of the worst-case scenarios. In which sense they are worst. From the point of hazard assessment view, some of them are not worst at all.**

Thanks, this is a good point, and we have added the following sentences:

For the short-term pre-eruption scenario we changed "worst case" to "large effusion rate" scenario (section 4.1.1) the first sentence now reads "The large effusion rate scenario was defined as fissures with an assumed effusion rate of 300 m$^3$/s which is considered high effusion rates on Reykjanes Peninsula."

And for the short-term scenarios described in 4.2.3 we added the following sentence in the 2$^{nd}$ section (L354-355)" The short-term worst-case scenarios were defined as a scenario where all lava emitted from the vent would be transported to one of the critical lava margins, e.g., lava margins close to a hiking path or infrastructure that could be inundated by the lava."

**- Line 272. Indicate the highways in Figs. 1 and 4 (black curve? or dashed curve?). Provide explanation of the curves in the figure captures.**

Thanks for pointing this out.

In the caption to figure 1 there is now a sentence describing this: "The highway is marked with a solid black line, and dirt roads with dashed black lines. Hiking paths are shown in dashed blue".

In figure 4 the following sentence has been added: **"**The highway is marked with a solid black line, and dirt roads with dashed black lines.**"**

**- Lines 294-95, 356-58, 368 (and elsewhere in the MS). "… during "emergency mode" response" (and similar sentences about meetings with the stakeholders). In Table 2, in column "Disadvantages" of the MrLavaLoba code, the authors write that "results not designed for hazard communication". If so, what do then the results of the numerical modelling mean for "emergency-mode" response? Clarify in the table and/or the text.**

Thanks, we have elaborated on this in Table 2 by writing: "Results not designed for hazard communication and templates to develop simple maps for hazard communication had to be produced."

So basically, it meant that we have to do extra processing steps in order to present the results from the numerical models. E.g., not present the lava flow thickness maps but just inundation areas, since this is the primary concern for the stakeholders (as explained in 4.2.3). Furthermore, we had to decide how to convey the uncertainty of the models (explained in 4.3).

When the templates were in place this was not a problem, but in the beginning the maps presented were not so easy for the stakeholders to understand.

**- Lines 374-75. "…the locations of critical lava margins were manually selected on the basis of the available knowledge and expert evaluation." Did the authors check the sensitivity of the manual selection on the inundation morphology?**

We do not fully understand this question, since it is unclear to us what the reviewer means by "inundation morphology". We have now (as you can see from the comment above regarding the worst-case scenario) explained better what defines the critical margins ")" The short-term worst-case scenarios were defined as a scenario where all lava emitted from the vent would be transported to one of the critical lava margins, e.g., lava margins close to a hiking path or infrastructure that could be inundated

by the lava." So, the manual selection was based on the available knowledge of the infrastructure and hiking paths. We did not make sensitivity analysis of this.

**- Lines 512-21. The first two paragraphs should be re-writen to avoid "copy-pasted" texts from abstract and the Introduction section.**
Thanks for pointing this out. Abstract, introduction and conclusion has been rewritten.

**- Table 1. First column "Vents". I guess that it is the number of vents (No. vents). Correct? If so, replace it. In the current line "Vents", please clarify the sign "/" (e.g., 2/10, Fissure 2/All… 5/Critical**
Thanks, the cell previously called "Vents" is now called "Description of vents" and "/" has been replaced by "or".

**In conclusion, addressing the issues here is crucial for the study to meet the standards of scientific rigour and contribute to the field in a meaningful manner. Once the clarity in organization of the paper and explanation/justification are provided, the result of the study can be considered relevant for publication in NHESS.**

**Alik Ismail-Zadeh**
**Karlsruhe Institute of Technology**